# Catastrophic impact of extreme 2019 Indonesian peatland fires on urban air quality and health
Mark J. Grosvenor [1,2,3] ✉, Vissia Ardiyani [3,4,5,6], Martin J. Wooster [1,2,3], Stefan Gillott[4,5], David C. Green [4,5], Puji Lestari [7] & Wiranda Suri [7]

Tropical peatland fires generate substantial quantities of airborne fine particulate matter ($PM_{2.5}$) and in Indonesia are intensified during El Niño-related drought leading to severe air quality impacts affecting local and distant populations. Limited in-situ data often necessitates reliance on air quality models, like that of the Copernicus Atmosphere Monitoring Service, whose accuracy in extreme conditions is not fully understood. Here we demonstrate how a network of low-cost sensors around Palangka Raya, Central Kalimantan during the 2019 fire season, quantified extreme air quality and city-scale variability. The data indicates relatively strong model performance. Health impacts are substantial with estimates of over 1200 excess deaths in the Palangka Raya region, over 3200 across Central Kalimantan and more than 87,000 nationwide in 2019 due to fire-induced $PM_{2.5}$ exposure. These findings highlight the need for urgent action to mitigate extreme fire events, including reducing fire use and landscape remediation to prevent peat fire ignition.

Landscape fires are common across much of southeast Asia, often used as a method for clearing and preparing land for new agricultural planting or other development[1-3]. Across the region, much of this activity takes place in previously cleared forest landscapes (e.g. scrub clearance fires) or on existing agricultural land (e.g. crop residue removal fires), as opposed to primary forest[4]. However, on the Indonesian islands of Kalimantan and Sumatra, vast areas of tropical peat underlies much of the previously cleared forest area and these carbon-rich soils are ignitable by surface fires and can burn for long periods when sufficiently dry[5,6]. Peat combustion can add very substantially to the surface fuel and carbon consumed in these fires[7-10]. By burning vertically downward as well as laterally, and in a way that mostly smoulders rather than flames, the amount of fuel consumed and particulate matter (PM) released per unit area burned in these fires is amongst the highest of any fire worldwide[11]. This ability to generate large amounts of airborne PM per unit area burned, coupled with the potentially extensive nature of the peatland fires even during climatologically relatively 'normal' years[12], can result in huge amounts of PM being released into the local atmosphere. The most extreme fires occur during times of El Niño-driven drought, where a combination of surface vegetation and forest fires along with burning peat soils can persist for many months[6,11,13-15].

During such extreme fires, the near absence of rain allows the PM released by the burning peat and surface vegetation to remain suspended in the air for long periods—generating enduring episodes of very poor air quality[16]. Locally termed 'haze', the polluted air is thick with toxic fine particulate matter ($PM_{2.5}$) capable of entering the human lungs and bloodstream[11,16-19], and this air pollution does not remain locally contained but rather can be transported by winds to affect areas hundreds of km away from the fires themselves[20,21]. During the most extreme fire episodes, this pollution can very significantly affect the air quality of nations surrounding Indonesia, and sometimes much of SE Asia itself[10,11,13,22,23].

The air quality hazard posed by this type of tropical peatland burning gained increased international attention after the El Niño exacerbated catastrophic fires of 1997, which were concentrated in the peatlands of Sumatra and Kalimantan[7,24,25]. This lead to the development of the Association of South East Asian Nations Agreement on Transboundary Air Pollution[26], though despite such policy developments, air quality modelling fed with satellite data on fire smoke emissions suggests that some of the world's worst air quality still results from the largest of these tropical peatland fire events[27-29].

[1]Department of Geography, School of Global Affairs, King's College London, London, UK. [2]NERC National Centre for Earth Observation, King's College London, London, UK. [3]Leverhulme Centre for Wildfire, Society and Environment, King's College London, London, UK. [4]Environmental Research Group, Analytical & Environmental Sciences, King's College London, London, UK. [5]Environmental Research Group, School of Public Health, Imperial College London, London, UK. [6]Nursing Department, Health Polytechnic of Palangka Raya, Palangka Raya, Indonesia. [7]Faculty of Civil and Environmental Engineering, Institute of Technology, Bandung, Indonesia. ✉e-mail: mark.grosvenor@kcl.ac.uk

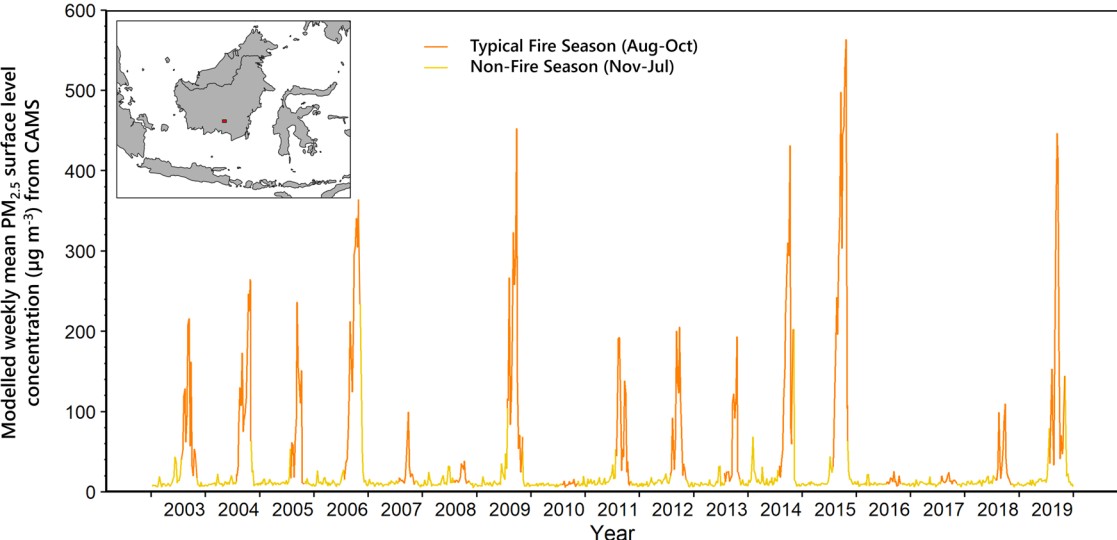

**Fig. 1 | Fine particulate matter (PM$_{2.5}$) concentrations modelled for Palangka Raya and the surrounding 1.5° × 1.5° area from data provided by the Copernicus Atmospheric Monitoring Service (CAMS).** These data are model-derived values, based on sources of PM$_{2.5}$ including satellite-derived biomass burning emissions and the CAMS atmospheric model. We have highlighted the August to October fire season each year from 2003 until 2019, when extreme increases in surface level PM$_{2.5}$ concentrations are routinely seen. Outside of these 3 months, PM$_{2.5}$ concentrations appear low (mean of 10 µg.m$^{-3}$)—demonstrating the generally very good air quality the region experiences for most of the year. The location of Palangka Raya is shown in the inset map. Data are from the CAMS EAC4 re-analysis dataset[33] and represent weekly means calculated across the four EAC4 0.75° (~75 km) grid cells containing Palangka Raya.

Figure 1 shows data on surface level PM$_{2.5}$ concentration for the area around Palangka Raya (Indonesia), the capital of Central Kalimantan and home to ~250,000 people. PM$_{2.5}$ is the most health impacting component of biomass burning smoke[30,31], and Palangka Raya is subject to very high PM$_{2.5}$ concentrations since it is located within a peatland area that often hosts fires during dry spells and droughts[13,32]. The PM$_{2.5}$ data shown in Fig. 1 are from model outputs provided by the Copernicus Atmospheric Monitoring Service (CAMS), which is led by the European Centre for Medium Range Weather Forecasts (ECWMF) to provide consistent, quality-controlled information on atmospheric composition and air pollution across the world, including near real-time estimates and forecasts of PM$_{2.5}$[33]. In areas of substantial fire activity, the CAMS forecasts are fed by satellite-derived data on fire emissions[9,34], and the types of extreme air quality reductions shown in Fig. 1 for Palangka Raya demonstrate the clear potential for considerable human health impacts from these peatland fire events[23,35–38]. However, whilst recent health impact studies have indicated the potential for such catastrophic impacts with ref. 29 calculating an average excess child (<5 year) death rate of between 19 and 38 thousand per year in Indonesia alone for example, the exposure to fire-sourced PM$_{2.5}$ used within these studies have mostly been based on large-scale modelled PM$_{2.5}$ concentrations of the sort shown in Fig. 1 (mean fire season PM$_{2.5}$ concentration of 97 µg.m$^{-3}$ (August-October) with background non-fire season mean of 14 µg . m$^{-3}$). Non-fire sources of PM$_{2.5}$ in the area may predominantly be from combustion engines in vehicles[13]. Because there are very few air quality measurements taken in areas such as Palangka Raya, whose air quality is suspected of being some of the most severely impacted by the peatland fire smoke, there is some uncertainty as to the efficacy of the modelled PM$_{2.5}$ concentration data in these severely haze-impacted regions. The AQ sensors that do exist are mostly located in more developed areas, such as Jakarta and Singapore, which are typically some hundreds of km away from the actual burning and thus experience probably far lower PM$_{2.5}$ concentrations than regions much closer to the fire source[39]. Here, we aim to improve this understanding of modelled PM$_{2.5}$ using a network of small sensors to assess air quality variability across the city.

To understand better what populations in areas like Palangka Raya are subjected to in terms of fire-sourced airborne PM$_{2.5}$, a small AQ network was installed around the city to quantify PM$_{2.5}$ concentrations during the 2019 extreme fire and haze episode, which was exacerbated by an El Niño.

Despite the El Niño itself being relatively weak, the modelled data shown in Fig. 1 suggests that the surface level PM$_{2.5}$ concentration around Palangka Raya was amongst the highest since the strong El Niño year of 2015. Our AQ network is the first to capture data within an urban area very close to these types of Indonesian peatland fires, and we use its data to both investigate the true urban air quality impact of the burning, and to evaluate the performance of the type of modelled PM$_{2.5}$ concentration data commonly used to support health impact assessments. If representative, earth observation derived estimates enable more continuous monitoring of AQ without the complications of sensor network characterisation and maintenance. Furthermore, reliable AQ data is a vital input when estimating health impacts within a region.

We focus our evaluation efforts on the state-of-the-art CAMS model whose data are shown in Fig. 1. These types of CAMS outputs have already been evaluated[40], but only in areas showing concentration ranges far below those expected during the type of extreme conditions seen in Kalimantan in 2019 (see Fig. 1 and ref. 11). Roberts and Wooster[41] have so far provided the only evaluation of CAMS data under more extreme fire-generated PM$_{2.5}$ conditions, but in that case mainly using individual PM$_{2.5}$ sensors located in e.g. American Embassies rather than from networks of sensors more densely placed within a fire affected region. Our sensor network was designed to provide data more representative of a CAMS grid cell than would be the case with data from a single point-based measure, and we hope to use these data to provide a guide to the confidence that can be put in the air quality and heath assessments stemming from use of these types of modelled PM$_{2.5}$ concentrations under life-threatening air quality conditions. Subsequent to this evaluation, we compare multi-year CAMS derived PM$_{2.5}$ surface level concentrations for Indonesian fire events to hospital derived patient data in order to demonstrate the gross-scale health impacts of the haze.

## Peatland fires in Kalimantan, Indonesia

Indonesia's carbon-rich tropical peatlands cover ~22,500 km$^2$ of mainly Sumatra and Kalimantan[38], representing ~8% of Indonesia's land area and 36% of the global tropical peatland area[42]. Widespread forest clearance and peatland drainage over many decades has made many of these Indonesian peatlands far drier and more flammable than previously was the case[43,44], and even during non-drought years surface fires can ignite the peat and reduce air quality for millions of citizens both in Indonesia and

neighbouring states[12]. In Kalimantan, which is the subject of this study, the annual fire season occurs between July and mid-November—peaking during September and October. However, drought in Indonesia often occurs during El Niño Southern Oscillation (ENSO) events, sometimes exacerbated by the Indian Ocean Dipole[27], and this brings further peatland drying and typically far more severe fire conditions[5,6,45,46]. ENSO-enhanced fire activity typically occurs on Kalimantan roughly every 3–5 years[1], with one of the most susceptible regions being Central Kalimantan, including the peatlands surrounding the capital city of Palangka Raya which is the largest city by land area in Indonesia (mean population density 93 per km$^2$ [47]). This region has been a prime location for commercial logging since the 1970's[48,49], and the 1990's 'Mega Rice Project' also cleared and drained huge areas for translocation-related rice agriculture[50,51]—ultimately largely being abandoned because, at least in part, the drained peat seemed not that well suited for the purpose. This landscape engineering has left behind a severely degraded and far more fire-prone terrain than the moist tropical forest that preceded it[48,51], and during the 2019 fires around 1.3% of the Palangka Raya district burned[52].

## Airborne particulate matter—measurement, modelling and health impacts
Surface level concentrations of airborne PM can be directly or indirectly measured[39,53–61]. They can also be estimated from modelling and/or remote sensing[8,10,23,28]. Each approach provides a powerful complementary method for estimating the magnitude and extent of poor air quality caused by airborne PM, both at individual locations and across regions and including different PM size fractions smaller and larger than PM$_{2.5}$[19,61,62].

Ground-based PM measurements must often seek a compromise between (i) collecting the most accurate direct 'reference' measurements of particulate mass per unit volume of air over a longer time period (e.g. 24 h), and (ii) providing higher temporal resolution measurements of particle number based on sensors using light scattering principles whose measurements are converted to PM mass per unit volume estimates using size fractions and assumptions of e.g. individual particle density[57]. For comparison to larger scale model-based PM concentration estimates, networks of small, low-cost PM sensors working on light scattering principles can offer advantages over the higher precision reference measurements - since it is cost effective to deploy many of them across a region. However, the sensor response to PM of the type being measured requires to be understood if the mass per unit volume data are to be considered accurate, which requires things like the particle density assumption to be well constrained[39,55–60,63]. Biased PM$_{2.5}$ concentrations can otherwise result[53,54], and even if such things are taken into account the influence of temperature, humidity, particulate concentration range and sensor degradation over time can also negatively affect the information generated[39,55–58,64–68]. Nevertheless, deployment of small low-cost sensors such as those reported in Roberts and Wooster[41] have resulted in explosion in the amount of data available to understand surface level PM$_{2.5}$ concentration variations, and thus the extent of human exposure to this harmful air pollutant.

Surface level PM$_{2.5}$ concentrations are also able to be modelled. And here we focus on modelled concentrations coming from CAMS EAC4 Reanalysis dataset[33,69], which are available from 2003 to the present. CAMS PM$_{2.5}$ data is from a forecast model[70] which uses a range of emissions inputs[71,72] including from fires[34]. Further description of the CAMS model is given in the 'Methods' section. In Central Kalimantan during the fire season, these emissions data show that almost all the airborne PM$_{2.5}$ is fire sourced, and measurements of air quality outside of the fire season also show the generally very good status of the breathable air in the region. We compare these CAMS PM$_{2.5}$ surface level concentration data to those from our PM sensor measurement network, providing a validation of the CAMS values at times of extreme El Niño enhanced fire activity.

In terms of health effects, breathing air laden with PM$_{2.5}$ is known to exacerbate risks of cardiovascular and respiratory disease and lung cancer[73–75], but most studies have focused on PM$_{2.5}$ from cigarettes[74] or urban sources[76]. However, recently ref. 31 have shown that PM$_{2.5}$ from

biomass burning appears even more toxic than that from non-fire sources, causing further concern for its effects on human health. The PM$_{2.5}$ from landscape fires can vary in its composition depending on the type of burning occurring. Peat fires are dominated by smouldering combustion and their emitted PM$_{2.5}$ by organic carbon, with a secondary black/elemental carbon component[11,14,77–79] and other compounds present in lesser quantities[22,77,78,80], whereas PM$_{2.5}$ from flaming vegetation fires typically has a greater black/elemental carbon component[78]. A global scale study by Roberts and Wooster[41] using CAMS outputs processed to focus on landscape fire-sourced PM$_{2.5}$ indicated that exposure resulted in ~678,000 premature deaths annually, ~39% of which are in the under-fives. Xue et al.[29] combined similar output from a different global air quality modelling approach with actual epidemiological data and information on child mortality to conclude that a 1 µg·m$^{-3}$ increment of fire-sourced PM$_{2.5}$ is associated with a 2.31% (95% CL 1.50–3.13) increased risk of mortality in the under-fives, providing an estimate of 28,900 (95% CL 19,100–38,400) excess child deaths per annum on average for Indonesia. The vast majority of these are related to the extensive tropical peatland burning in Kalimantan and Sumatra. Focusing on Central Kalimantan alone, Uda et al.[38] estimated that on average an additional 648 deaths occur per year (around 4.4% of total deaths) as a result of exposure to fire sourced PM$_{2.5}$ (2011–2015). However, the accuracy of the surface level PM$_{2.5}$ estimates coming from the air quality models used in these works is largely unknown for the type of extreme haze situation found during El Niño exacerbated fire events, supporting our intended effort to use measurements from our PM$_{2.5}$ sensor network to evaluate this.

## Specifics of air quality and health impacts in central Kalimantan
The location of Palangka Raya is shown in Fig. 1, and outside of the Aug to Oct low-rainfall period the region is characterised by typically very good air quality (mean PM$_{2.5}$ = 10 µg·m$^{-3}$; 2015–2019, CAMS EAC4 dataset[33]). Between August and October, annually recurrent peatland fires close to or even within the city boundaries can lead to frequent air quality reductions[27] however, in 2019 the local fire season was particularly severe, as it was in many other parts of Kalimantan and Sumatra. Despite the forecasts of a strong El Niño and an extreme drought not actually transpiring[81], a relatively weak El Niño did occur but ceased by August 2019[82]. Nevertheless, the peatland fires of 2019 were the largest documented since the strong El Niño year of 2015[6,11] and CAMS outputs confirm that weekly mean surface level PM$_{2.5}$ concentrations also reached a modelled maxima second only to that experienced during 2015 (Fig. 1).

In terms of health at the local scale, in addition to the direct effects on the population breathing the smoke, it is known that exposure to PM$_{2.5}$ by pregnant women can lead to low birth weights and further health issues for the children[83,84] and can also lead to chronic non-communicable diseased in adult life[85,86]. Data from hospitals in the city of Palangka Raya shown in Supplementary Fig. 1 indicate a relationship between birth weight and potential PM$_{2.5}$ exposure, with the latter derived from the CAMS surface level PM$_{2.5}$ concentration data shown in Fig. 1. This suggests that exposure of pregnant women in Palangka Raya to the fire-sourced PM$_{2.5}$, which happens on a regular basis (occurring in 70% of years, Fig. 1), is having a detrimental impact on their babies. However, in order to investigate such potential relationships further, it is important to have confidence in the accuracy of the particulate exposure datasets used—which is one aim of the current study.

## Results and discussion
### Land cover and fire activity
Figure 2a highlights the installation locations of the air quality sensors installed in the city of Palangka Raya during the 2019 fire season. Figure 2b indicates that the fire radiative power (FRP) of the vast majority of the active fires detected in the area is very low, and is probably associated mostly with smouldering fires in the peat surrounding the city. With these types of fires, whilst some above-ground biomass is consumed with flaming combustion, the smouldering peat combustion with lower surface-emitted FRP[15] continues for longer periods and therefore is more likely to be detected by polar-

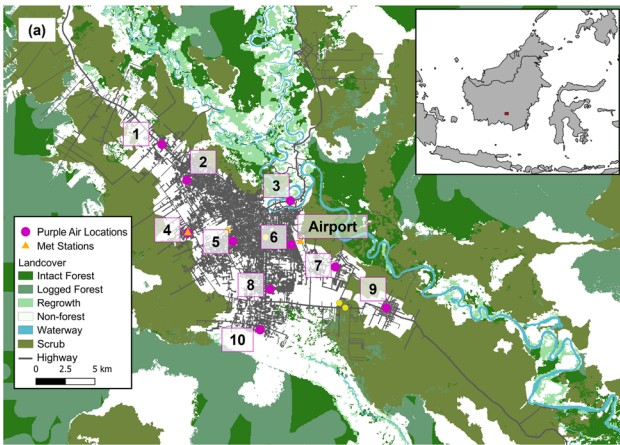

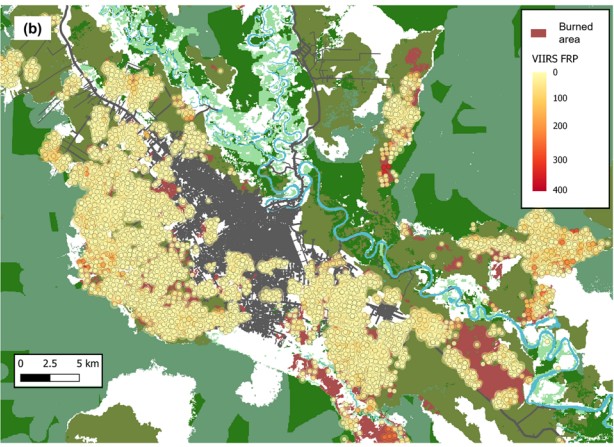

**Fig. 2 | Landcover and hotspots detections in Palangka Raya. a** Superimposed is each location where a low-cost 'Purple Air' particulate matter (PM) measurement device was deployed. A meteorological station was also present alongside the PM measurement location 4 (Jekan Raya Borough Office), and the location of the Palangka Raya Airport meteorological station is also shown since it served as the location for the sensor co-location tests. Landcover data is based on that from ref. 87.

**b** Map of active fire hotspots detected around Palangka Raya made using data from the spaceborne Visible Infrared Imaging Radiometer Suite (VIIRS)[107,108]. Colour of hotspot shows its FRP. Also shown is the 2019 burned area which extends beyond the area of hotspot detections[109]. Basemap features are from OpenStreetMap (openstreetmap.org/copyright).

orbiting satellites. Indeed, observations in September 2019 around the region found almost no flaming activity ongoing, but huge amounts of smouldering peat (as was described in the same area in 2015 by Wooster et al.[11]. According to the landcover map produced by Gaveau et al.[87], within a 100 km radius of Palangka Raya city centre, 64% of the VIIRS-detected AF hotspots occur on scrubland, 29% on other 'non-forest' land. Only 5% are located on logged forest, 2% on intact forest, and 0.5% on regrowth forest. It is noted however, that there is likely to have been some degree of land cover change between 2016 when the land cover map was published and the 2019 fire season. What is particularly notable from Fig. 2b, is the proximity of hotspots to the city—with many occurring within the city boundary and therefore very close to where the population live and work. It is also notable that areas of protected and far less degraded land, such as the Sebangau National Park region to the southwest of the city, are almost entirely free of hotspots. Further discussion of these active fire data including relationship with distance from the city, and pattern of total FRP and hotspot count is given in Supplementary Notes 1 and Supplementary Fig. 2.

### PM sensor co-location and calibration results

Figure 3 shows examples of data taken from the pre-deployment co-location tests of the Purple Air (PA) sensors subsequently used to form the PM sensor network around Palangka Raya. These tests were conducted at Palangka Raya airport, where the fires were some distance away and the landscape open and not intruded upon by tall buildings and high vegetation —leading to relatively spatially uniform PM concentrations in the test location. The generally good agreement between the $PM_{2.5}$ data of each PA sensor is apparent, and statistics derived from the two pre-deployment tests conducted between 15 and 18 August 2019 and from post-deployment test (27 October 2019) are shown in Table 1. Each PA sensors contains two Plantower light-scattering PM detectors, and the table reports the percentage mass concentration difference between the two, along with the mean of that from all. Whilst the mean percentage difference of all detectors is low (1.4%), a few showed differences exceeding 10%. Where such large differences existed between detector A and B of the same PA sensor—in the data from the Palangka Raya deployment the data from the detector that best matched the overall mean in Table 1 was used (rather than the mean from both detectors). In this case, differences always remained below 2.4%. The table also indicates that no clear drift in PA sensor performance occurred between the pre-deployment and post-deployment periods, providing confidence in the stability of the Plantower detectors despite their exposure

to very high PM concentrations during the intervening fire season, including concentrations beyond what they are designed to be able to measure.

As already stated, the Plantower detectors present within the Purple Air devices measure light backscattering from airborne particulates, as do many other PM sensors such as the TSI DustTrak[11]. These light scattering measures are converted to PM mass concentrations using assumptions appropriate for a particular type of PM[56,65], but these assumptions may not fit with the peatland fire smoke PM source focused on herein. The PM source in Palangka Raya during the fire season is totally dominated by smoke from peatland burning, and comparison of the PA sensor output collected during the co-location tests to contemporaneous gravimetric PM reference measurements made using a MiniVol reference sampler[88] showed the need for an adjustment factor (AF)[63]. This study advanced previous work by Wooster et al.[11] who derived an AF for this environment using a TSI DustTrak during the 2015 fires surrounding Palangka Raya. Other AFs for Purple Air (Plantower) sensors in different situations have been made by Delp and Singer[58], Malings et al.[66], Mehadi et al.[67] and Tryner et al.[60], however, it was necessary for this study to derive a new AF specific for Purple Air (Plantower) sensors being used in smoke from tropical peat. Combining these gravimetric PM measurements with data from Purple Airs ID.3 and ID.5 co-located with the MiniVol at the locations of fires on several days during the peak of the fire season, as well as during Co-location Tests 1 and 2, provides an updated AF relevant to the PA sensors used herein. Figure 3b shows the derivation of this 0.49 AF, which is in fact almost identical to that derived for the TSI DustTrak by Wooster et al.[11]. Whilst reference samplers like the MiniVol provide very useful gravimetric measures of the mean $PM_{2.5}$ concentrations found over a 24 h period, data from the Purple Air sensors shown in Fig. 3a indicate the substantial concentration variability that is seen over this time period. Over 24-h, the Purple Airs may in fact be exposed to periods of such high $PM_{2.5}$ concentration that it is above the design limits of the Plantower detectors (effective range: 0–500 µg . m⁻³, maximum range 1000 µg . m⁻³ [58]), whilst at other times lower concentrations are well within these. To gain better knowledge of how the highest concentrations may affect the overall $PM_{2.5}$ dataset, we conducted a series of combustion chamber calibration tests. These are detailed in Supplementary Notes 2 and Supplementary Figs. 3, 4), and highlight a high concentration adjustment of *0.67x-225* which is applied to the 2.7% of data points measured by the PA sensors at >1000 µg . m⁻³. This finding demonstrates that Plantower sensors can be used in extreme environments to quantify air quality.

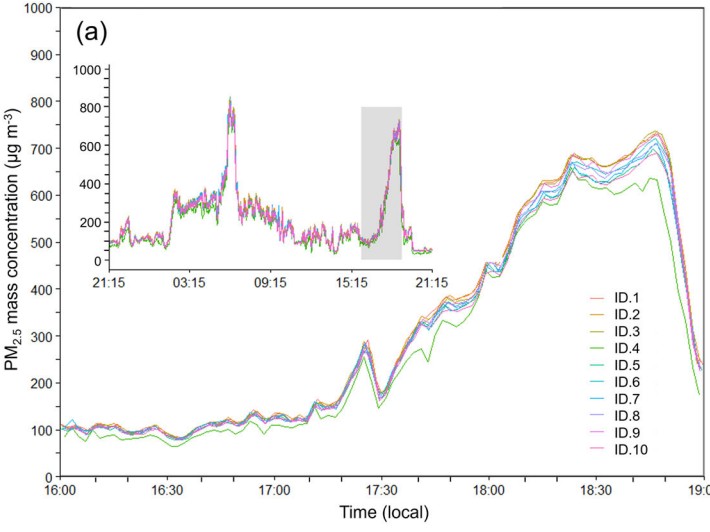
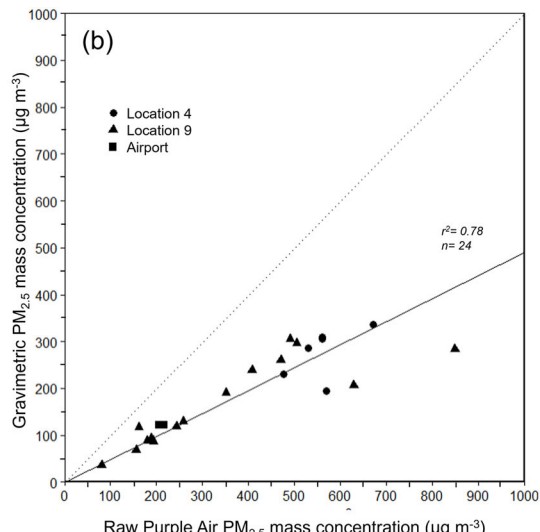

**Fig. 3 | Performance of Purple Air sensors in Palangka Raya. a** Co-location of 10 Purple Air sensors. Y-axis shows raw Purple Air mass concentration. Each device contains two Plantower PM detectors (A and B) and this example is from one of the two pre-deployment tests (Test 2; Table 1) conducted on 17–18th August 2019 at the BMKG meteorological station location at Palangka Raya Airport whose location is shown in Fig. 2. Values shown are the mean of those from each Purple Air Plantower 'A' and 'B' detector. While insert shows the full 24-h time-series, the main plot shows the detail of a 3-h period during which a strong $PM_{2.5}$ concentration peak occurred (grey box on insert). Subsequent to this co-location period each device was re-located to its August-October deployment postion shown in Fig. 2. **b** Mass correction of Purple Air sensors against gravimetric filter samples using a MiniVol instrument with the linear regression slope of 0.49 (intercept = 0). Samples were collected from 3 locations in Palangka Raya.

## PM data from city-wide deployment

Subsequent to the co-location tests, the Purple Air sensors were deployed across Palangka Raya in the locations shown in Fig. 2 and most operated almost continuously for the 2 month period between 20th August and 22nd October 2019 without issue. On average each detector on each PA sensor captured data for more than 93% of its deployed period. Gaps in the dataset are most likely caused by temporary issues with the power supply to the sensors. Figure 4a shows 3-h averages of the $PM_{2.5}$ data collected by each sensor, with the AF detailed above already having been applied. While the overall temporal pattern is similar across all sensors, there is a substantial amount of inter-site variability - with the highest concentrations occurring

14–16th September when 3-h averages exceeded $700\,\mu g \cdot m^{-3}$ at all locations. Differences seen in the concentrations recorded simultaneously at different locations is likely due to a combination of fire source location and wind direction, with some potential influence from the immediate environment close to each PA device—such as the presence of trees or buildings that may help remove $PM_{2.5}$ from the air[89] alongside the possibility of some localised non-fire sources contributing to the individual sensor measurements. Figure 4b shows data from a single PA sensor (ID.4) which was co-located with a fixed air quality monitoring station (AQMS), although at the start of the fire season the latter was non-operational (and it was calibrated following the fire season, so data may not be typical of this instrument). There is broad agreement in the trends shown by the AQMS and co-located PA, although the AQMS shows far greater peak concentrations possibly highlighting a need for an appropriate AF during times when $PM_{2.5}$ is dominated by smoke sources rather than road dust and fossil fuel emissions. Figure 4b also shows daily precipitation totals for the city—highlighting the total lack of rain between 2nd and 19th September where airborne $PM_{2.5}$ concentrations reach a peak. Consistent rainfall does not occur until 12th October onwards, when $PM_{2.5}$ concentrations are already substantially reduced compared to the peak period.

Across the city, the PA data of Fig. 4 indicates that mean $PM_{2.5}$ mass concentration between 20th August and 24th October is $137\,\mu g \cdot m^{-3}$. World Health Organisation (WHO) 24-h mean air quality guidelines at the time of exposure recommended a maximum of $25\,\mu g \cdot m^{-3}$ [90], clearly indicating the extreme nature of the air pollution being experienced across the city. This limit has more recently been revised downwards to $15\,\mu g \cdot m^{-3}$ reflecting increased knowledge of the link between $PM_{2.5}$ and health impacts[91]. The CAMS modelled $PM_{2.5}$ concentration data discussed later (Fig. 7) indicate that the PA deployment captured the bulk of the severe air pollution event. Prior to the sensor installation, a short period of haze between 4 and 18th August was not measured.. Nevertheless, even if $PM_{2.5}$ mass concentrations in the breathable air fell to near zero outside the fire season, the PA time-series collected during the 3-month fire season indicates that annual $PM_{2.5}$ exposure for the populace of Palangka Raya in 2019 is likely to have been more than double the WHO annual mean guidelines ($10\,\mu g \cdot m^{-3}$)[90]. The PA sensors also reveal the diurnal nature of the air pollution generated by the peatland burning (Fig. 5), with $PM_{2.5}$

## Table 1 | Statistics derived from the three Purple Air co-location tests

| Purple Air Sensor ID | Plantower Detector 'A' | | | Plantower Detector 'B' | | |
|---|---|---|---|---|---|---|
| | Aug | | Oct | Aug | | Oct |
| | Test 1 | Test 2 | Test 3 | Test 1 | Test 2 | Test 3 |
| ID.1 | 10.3 | 12.0 | 11.3 | −1.7 | −1.0 | 0.1 |
| ID.2 | 12.6 | 14.1 | NA | −1.8 | −1.1 | NA |
| ID.3 | 1.9 | 2.8 | 3.2 | 5.8 | 6.6 | −1.7 |
| ID.4 | −3.7 | −5.7 | 3.7 | 3.9 | −6.0 | 11.9 |
| ID.5 | −2.4 | −2.7 | NA | 2.1 | 0.8 | NA |
| ID.6 | −0.1 | 1.3 | NA | 1.1 | 2.3 | NA |
| ID.7 | −3.5 | 2.5 | −6.8 | 1.4 | 3.3 | −2.8 |
| ID.8 | −1.1 | 0.1 | −1.0 | −0.4 | 1.4 | 2.8 |
| ID.9 | 2.3 | 2.1 | −1.6 | 1.2 | 1.7 | 0.9 |
| ID.10 | −1.2 | −0.2 | −2.0 | −3.1 | −1.9 | −0.9 |

Test 1 and 2 occurred 15–19th August 2019 prior to sensor deployment around Palangka Raya, and Test 3 occurred on 27th October 2019 after the fire season ended and the sensors were removed from their deployment locations shown in Fig. 2. Each number in the Table represents the % difference between the $PM_{2.5}$ concentration measurement delivered by each of the two (A and B) Plantower PM detectors present in each Purple Air sensor (ID.1 to ID.10) compared to the mean of all used in the test.

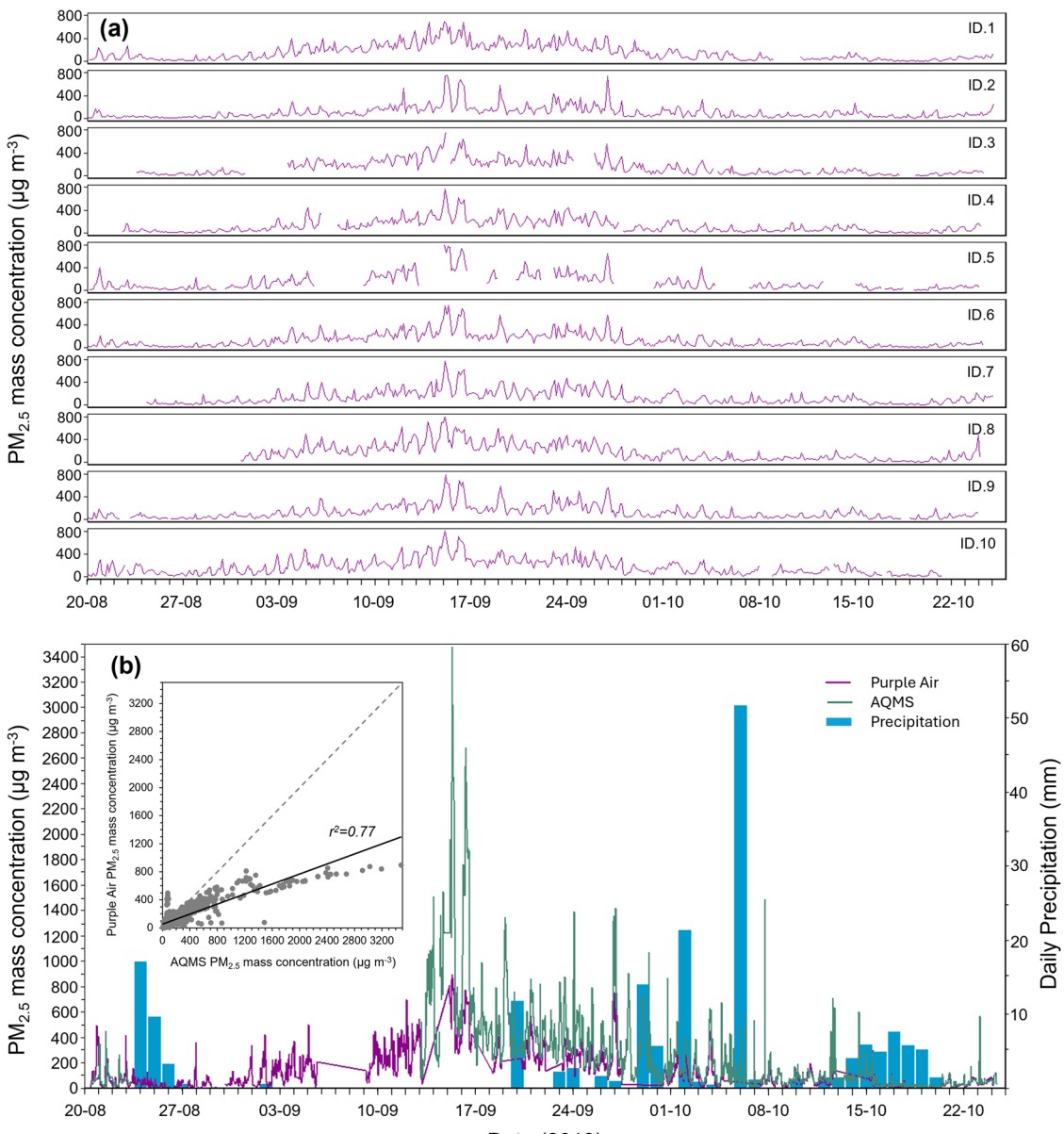

**Fig. 4 | Air quality and precipitation in Palangka Raya. a** Fire season $PM_{2.5}$ mass concentration time series taken from the ten Purple Air sensors deployed across the city as shown in Fig. 2a, and (**b**) 30-min $PM_{2.5}$ mass concentration time-series from the single Purple Air sensor (ID.4) co-located with a fixed air quality monitoring station (AQMS) at the Jekan Raya Borough Office in Palangka Raya. Daily precipitation data is also shown (CPC Global Unified Data, Physical Sciences Laboratory) in (**b**), along with the insert plot showing the best fit linear relationship between the Purple Air and AQMS data (30 min mean values).

concentrations typically reaching their daily maximum in the early morning (peaking ~ 06:00 h local time) despite peatland fires typically reaching their peak in the afternoon or early evening (see data in refs. 11,46). After the early morning peak in air pollution, a daytime decline and a late afternoon rise occurs, reaching a secondary peak around 17:00 h local time before a slight temporary reduction and a rise again from around 20:00 h. The early morning peak in $PM_{2.5}$ mass concentration is likely a direct result of the overnight shrinkage of the atmospheric boundary layer, which aids trapping of the air pollutants at night and their increased dissipation during the day as the boundary layer once again extends upwards[92,93]. This effect may even be exacerbated by enhanced stability of the boundary layer related to the extreme $PM_{2.5}$ loading[94].

### Air quality metrics
Air quality data are often expressed to the public via air quality indices, rather than concentrations of individual species such as $PM_{2.5}$. One of the most commonly used indices is the US EPA's Air Quality Index (AQI), originally named the Pollutant Standards Index[95]. We used the 24-h mean $PM_{2.5}$ mass concentrations coming from our Purple Air network to derive AQI measures, though it should be noted that the Indonesian national AQ reporting system uses instead $PM_{10}$ measures, since most local measurement stations available at the time reported this size fraction rather than $PM_{2.5}$. Data from the Purple Air sensors show that 90% of PM within the $PM_{10}$ classification falls within the $PM_{2.5}$ fraction although there is a degree of variability ($\sigma = 2.7$). Nevertheless, since the finer particulates are those most relevant to human health[80] we focus on this size fraction, with our resulting AQI data are presented in Fig. 6 and summarised in Table 2. There are several days where the AQI for Palangka Raya exceeds even the maximum 'extreme' value of 500, and only for 1 day during the August to October 2019 fire season was Palangka Raya classified as having 'good' air quality. Eight days were classified as hazardous (6 days as H1, 2 as H2), and the majority as unhealthy (30 days) or very unhealthy (15 days).

**Fig. 5 | Diurnal variability of PM$_{2.5}$ concentrations in Palangka Raya.** Daily average 30-min mean PM$_{2.5}$ mass concentrations as measured by the ten Purple Air sensors deployed at the locations shown in Fig. 2. Means are calculated from data taken over the entire deployment period (20 August–24 October 2019).

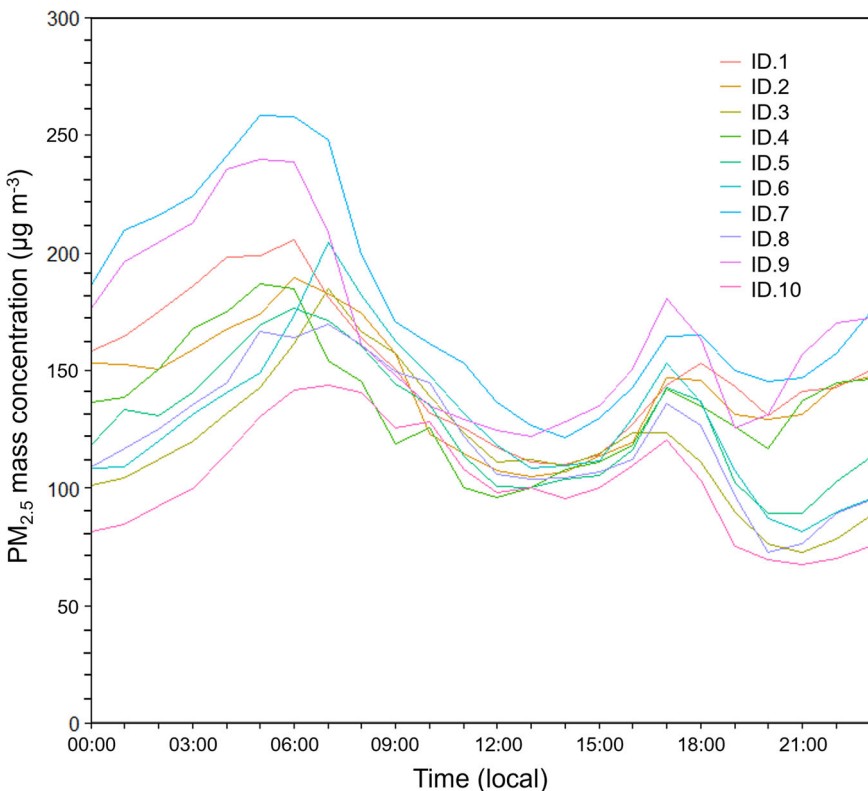

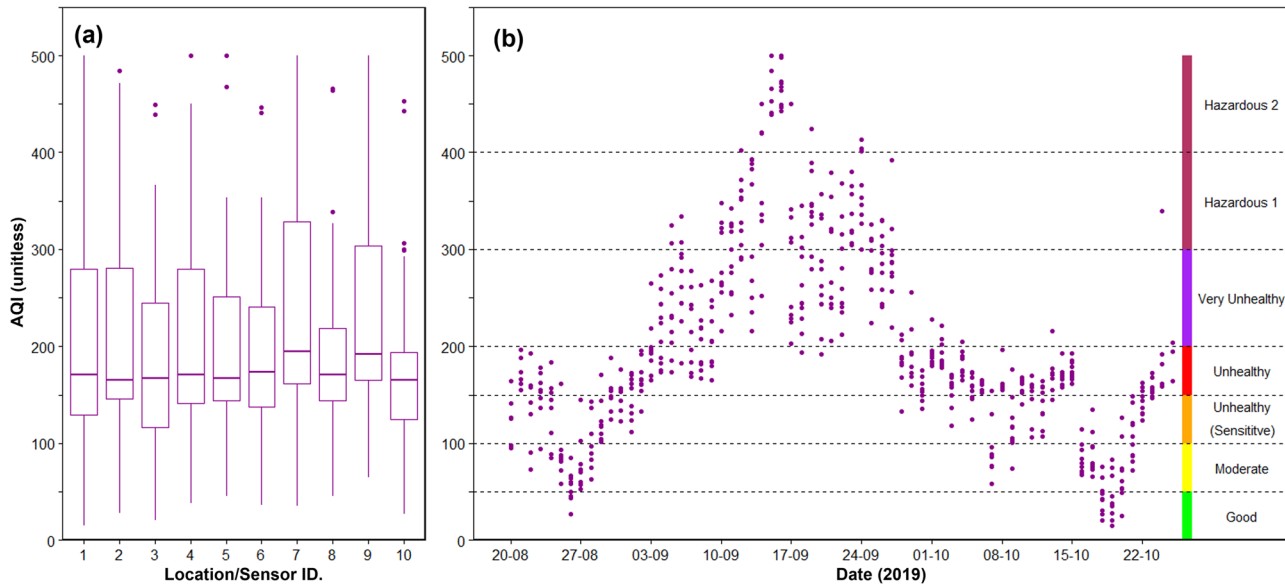

**Fig. 6 | Air Quality Index during 2019 fire season in Palangka Raya. a** Boxplots showing EPA AQI calculation for each location based on 24-h average of airborne PM$_{2.5}$ mass concentrations (thick bars define median, box shows 25th-75th percentiles, whiskers are 1.5x interquartile range, points show outliers) and (**b**) time series of daily AQI calculated from those concentrations. Points are displayed for each of the 10 Purple Air measurement locations shown in Fig. 2.

The differing PM$_{2.5}$ concentrations seen at locations across the city (Fig. 4a) resulted in substantial AQI variability between sites, as can be deduced from the data of Table 2 and Fig. 6a. One site (Location 7, Tanjung Pinang) experienced 20 days of 'Hazardous' AQ conditions for example, whilst another only 3 days (Location 10, Kereng) (Table 2). Conditions classed as 'Good' or 'Moderate' existed for between 5 (Location 9, Kalampangan) and 13 days (Location 1, PHC Jekan Raya). Knowledge about this degree of variability across a city is valuable as this may be missed when using modelled (earth observation) estimates, or when relying on a limited

number of fixed monitoring stations. Evidence that small low-cost sensors such as these are capable of delivering a rich dataset that could enable improved response (from, e.g. local government) to those areas having worst air quality. There is some evidence to suggest that the more severe air quality existed at sites furthest from the city centre, and the site with best air quality (Location 10, Kereng) is located to the very south of the city - closer to areas of wetlands and protected forest (Fig. 2). Forest areas typically have the ability to remove particulates from the air via their interception by leaves and other vegetation components[96], and during the 2015 fieldwork described in

**Table 2 | Air Quality Index classifications. Number of days in each classification by site**

| Location | | Good | Moderate | Unhealthy (sensitive groups) | Unhealthy | Very unhealthy | Hazardous 1 | Hazardous 2 | Extreme | Total days assessed |
|---|---|---|---|---|---|---|---|---|---|---|
| ID.1 | PHC Jekan Raya | 3 | 10 | 6 | 21 | 11 | 11 | 3 | 1 | 66 |
| ID.2 | MC Hiu Putih | 3 | 8 | 5 | 18 | 14 | 9 | 3 | 0 | 60 |
| ID.3 | MC Pahandut | 3 | 9 | 9 | 22 | 12 | 6 | 2 | 0 | 63 |
| ID.4 | BO Jekan Raya | 2 | 5 | 11 | 19 | 8 | 9 | 1 | 2 | 57 |
| ID.5 | HH G. Obos | 2 | 9 | 8 | 23 | 17 | 5 | 1 | 1 | 66 |
| ID.6 | MC Ramin | 2 | 7 | 9 | 23 | 14 | 6 | 2 | 0 | 63 |
| ID.7 | PHC Tanjung Pinang | 2 | 4 | 4 | 19 | 6 | 14 | 4 | 2 | 55 |
| ID.8 | PHC Marina | 2 | 7 | 11 | 28 | 11 | 6 | 2 | 0 | 67 |
| ID.9 | PHC Kalampangan | 0 | 5 | 5 | 25 | 11 | 13 | 3 | 1 | 63 |
| ID.10 | PHC Kereng | 1 | 11 | 15 | 25 | 13 | 1 | 2 | 0 | 68 |
| City (mean of all sites) | | 1 | 9 | 5 | 30 | 15 | 6 | 2 | 0 | |

Extreme category is where AQI > 500 (2h-hour mean $PM_{2.5}$ > 500 μg . m$^{-3}$).
Location name abbreviations are as follows: *PHC* Primary Health Centre, *MC* Midwifery Centre, *BO* Borough Office, *HH* household.

ref. 11 when $PM_{2.5}$ concentrations in Palangka Raya were at least as extreme as in 2019 and probably worse (see Fig. 1) a visit to Sebangau National Park to the southwest of the city revealed an area with visibly less airborne $PM_{2.5}$ (see Supplementary Fig. 5). Furthermore, these protected areas are often prioritised for protection from fires through helicopter water drops.

At Location 4, the Purple Air was located very close to the AQMS, such that their $PM_{2.5}$ data can be compared (Fig. 4). Though the temporal pattern reported by both instruments is similar, the AQMS appears to deliver consistently higher $PM_{2.5}$ mass concentrations, raising the possibility that the AQMS data itself requires an AF appropriate to peat fires. This seems quite likely, as these type of instruments usually target urban $PM_{2.5}$ sources—which are dominant in Palangka Raya during the periods outside of the fire season.

**Comparison of Palangka Raya measured and modelled surface level $PM_{2.5}$ data**

For the 2019 fire season, we compared the measured $PM_{2.5}$ mass concentration data coming from our Palangka Raya-based network of Purple Air sensors (Fig. 4a) to the simulated concentrations coming from the CAMS EAC4 model (Fig. 1). As detailed previously, the CAMS surface level $PM_{2.5}$ concentration estimates are described in ref. 33 and are produced at 3 h intervals on a 0.75° grid (equivalent to $84 \times 84$ km in Indonesia). Palangka Raya sits at the intersection of four CAMS grid cells, and we, therefore, compared the mean modelled $PM_{2.5}$ mass concentration in these four grid cells to the city-averaged $PM_{2.5}$ value taken in situ using our Purple Air sensors (Fig. 7). It is noted that this larger area of earth observation analysis means that the city area forms a smaller proportion of the cells (9.4%) and therefore the CAMS data will also consider AQ from source areas beyond the city area, however, this type of situation may be typical of many locations where analysis of a single cell is not representative of what that location is subjected to.

There is generally rather good agreement shown between the in situ and CAMS datasets, with a very similar temporal development ($r^2 = 0.80$) and relatively good agreement in magnitude—especially at the lower $PM_{2.5}$ concentrations seen before and after the September 2019 air pollution peak. In September, when daily average $PM_{2.5}$ mass concentrations increased well above 100 μg . m$^{-3}$, there are larger differences shown between the in situ and modelled data—with the latter suggesting somewhat higher concentration peaks. For days when city-wide means exceed 100 μg.m$^{-3}$ according to the PA data, the CAMS EAC4 model provides higher values by a mean of 35%, whilst for days when the measured concentration was <100 μg . m$^{-3}$ they are far closer (on average 5% lower). Christophe et al.[40] previously evaluated CAMS' surface level $PM_{2.5}$ mass concentrations between March and May 2019, comparing them to in situ records from 160 'background sites' across Europe and North America. They found CAMS estimates to be on average around 54% higher than the in situ measures in North America, but 17% lower in Europe. Roberts and Wooster[41] focusing more on fire affected areas found that CAMS data both over and under-estimated $PM_{2.5}$ concentrations depending on the year, by up to 56% and 38% respectively, and overall overestimated surface level $PM_{2.5}$ mass concentrations by on average by 24%. Therefore, the agreement we see between CAMS and our in situ $PM_{2.5}$ data from Palangka Raya appears to be of similar strength to these prior studies, and in some ways rather better. The greater differences we find during the September period of peak fire and air pollution is highly likely to be contributed to by the fact that the CAMS dataset covers a much wider area than just the city of Palangka Raya where the Purple Air sensors were located, and indeed includes the surrounding peatlands where fires were burning at this time right on the edge of the city (see Fig. 2b). $PM_{2.5}$ concentrations will likely be maximised close to fire sources compared to the city (see Supplementary Figs. 2, 5), and those in the smoke plumes themselves can reach 20,000 μg . m$^{-3}$ or higher (see ref. 11), more than an order of magnitude more than in the city itself. MiniVol filter data collected from source locations in September 2019 show concentrations between ~600–2700 μg . m$^{-3}$. In October, when fires had generally moved further from the city (see Supplementary Fig. 2) but where it was still experiencing the impacts of smoke carried by distant plumes, the modelled and in situ measured concentrations match rather well ($r^2 = 0.85$ 9–24th October).

Whilst the temporal pattern of the measured and modelled $PM_{2.5}$ surface level concentration data shown in Fig. 7 shows relatively good agreement, a key consideration affecting the detail is whether or not the wind is blowing smoke from fires into the city or away from it. The highest $PM_{2.5}$ surface level concentration present in the CAMS data is that on 26th September 2019 (Fig. 7), but this peak is not apparent in the in situ data for example. This may be related to the fact that at this time the strongest fire activity was located to the north-east of Palangka Raya (see the active fire detections detailed in Supplementary Notes 1 and Supplementary Fig. 2), starting on the 23rd and continuing through the 26th September, whilst the wind direction was from the south-east. Thus the smoke from these fires would not influence the in situ measurements made by our Purple Air network, though it would influence the CAMS modelled $PM_{2.5}$ data since these fires do fall within the boundaries of the CAMS grid cells that include Palangka Raya.

In addition to our analysis with the CAMS EAC4 dataset, we also investigated the agreement between the in situ $PM_{2.5}$ data and that from the CAMS Near Real Time (NRT) service, which is generated at a higher spatial resolution (0.4° grid) than the EAC4 model[40]. The NRT model has the advantage that it is produced on a shorter time scale meaning it could be

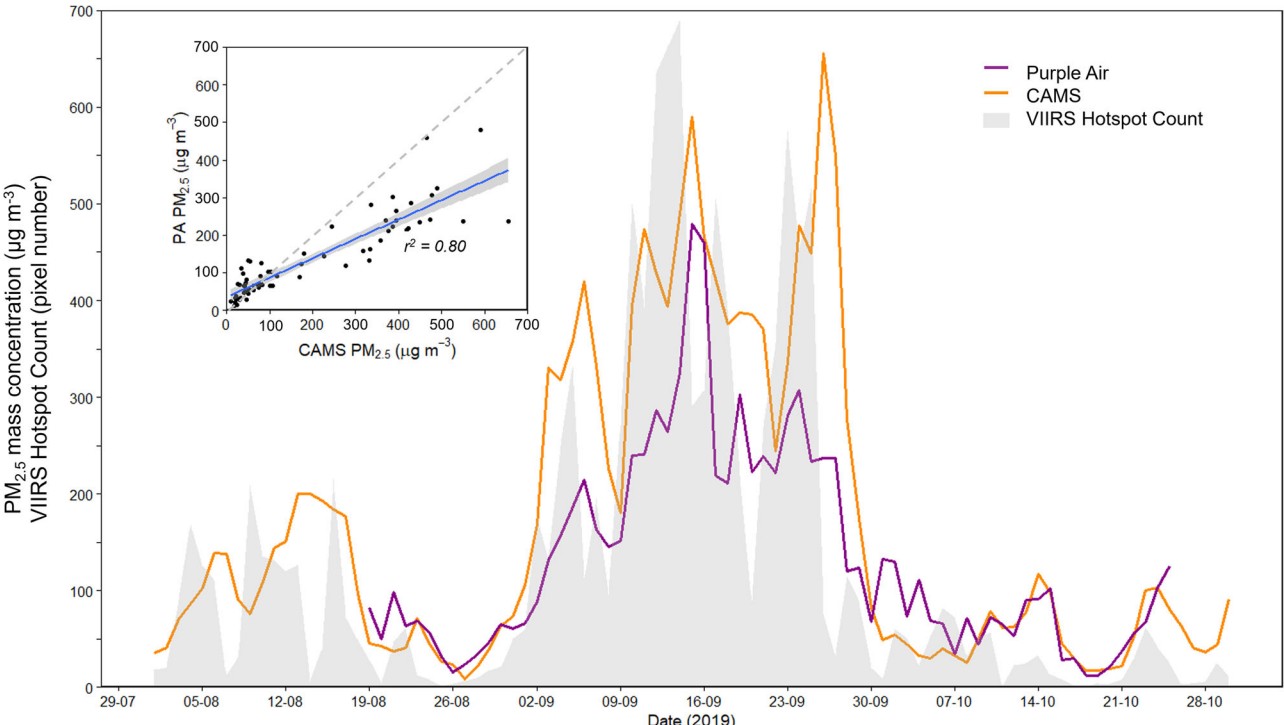

**Fig. 7 | Time-series of measured (Purple Air) and modelled (CAMS) PM$_{2.5}$ mass concentration data, displayed together with the record of VIIRS active fire pixels detected in the same area.** Time-series of daily mean city-wide airborne PM$_{2.5}$ concentrations taken from the ten Purple Air sensors deployed during the 2019 fire season, as shown in Fig. 2, alongside the same metric calculated from CAMS outputs for the four 0.75° grid cells covered by Palangka Raya. The grey shaded area represents the daily number of active fire pixels detected by VIIRS.

more useful for decision making, however, the EAC4 reanalysis model benefits from additional inputs which should result in higher accuracy[40]. To enable comparison to the EAC4 gridded data, we calculated the mean of nine NRT grid cells to cover as similar an area as possible to the EAC4 grid (1.2° compared with 1.5°). The coefficient of variation between the NRT model output and our in situ Purple Air measurements is slightly reduced compared to that between the EAC4 and the PA data ($r^2 = 0.70$ as opposed to $r^2 = 0.80$). On days when the in situ measurements were >100 μg . m$^{-3}$, the NRT dataset was lower than the in situ data by on average 66%, whilst on other days it was lower by 45%. However, it is important to acknowledge the global variability of these differences when seeking to understand estimations for one particular region and in particular how these biases may be influenced by concentration. It is possible that the estimates from the NRT dataset are poorer than the EAC4 dataset, partly due to their higher resolution grid over a region with relatively diverse ground conditions, and with the variable measures within the city area depending not only on the level of fire activity but also its location (which may be clustered) and wind direction. Where fire activity and wind direction do not move smoke over the city, there will likely be increased divergence between what is measured by the in situ Purple Air network and what is estimated by the CAMS model. This is a key limitation to the analysis in this study using a large scale model to understand more localised impacts where within each cell there is a high degree of variability of fire conditions, and meteorological factors affecting its representativeness.

**Health impact assessment**

Exposure to the type of fine particulate matter (PM$_{2.5}$) focused on herein represents the greatest ambient air pollution risk to health, and in the most extreme cases can result in the mortality of individuals. Following Roberts and Wooster[41], we used the method of Johnston et al.[97] to estimate the excess mortality resulting from chronic exposure to the elevated PM$_{2.5}$ concentrations seen in Palangka Raya. Details of the equation used is given in the 'Methods' section.

Whilst many methods used to estimate attributable mortality have been developed to work on a global or regional basis[23,35–37,97,98], we applied this relation to the region around Palangka Raya (1.5° × 1.5°). For Palangka Raya and its surroundings, the counterfactual PM$_{2.5}$ concentration is zero, since all fires are ignited by human activity, whilst the PM concentrations were taken from the CAMS outputs of the same 4 grid cells as analysed in Fig. 7. Background non-fire PM$_{2.5}$ concentrations are calculated based on non-fire season concentrations to account for exposure to non-fire sources of PM$_{2.5}$ (such as vehicle emissions). Population was estimated from a 1 km resolution gridded population map adjusted for UN methodology[99], and all-cause mortality from World Bank estimates for Indonesia[100].

Results (Table 3) show that in extreme fire years such as 2015 and 2019 there are more than one thousand attributable deaths in the wider Palangka Raya region (pop. 679,000) as a direct result of landscape fire PM$_{2.5}$ exposure. For comparison, the World Bank estimates for Indonesia suggest around 4000 all-cause deaths occur annually in the same region. Expanding our method using CAMS EAC4 data from across Central Kalimantan (pop. 2.9 M) and across Sumatra and Kalimantan as a whole (74.7 M) as a whole we estimate excess deaths from landscape fire PM$_{2.5}$ exposure in 2019 to be 3276 and 51,377 respectively, which compares to 4910 attributable deaths in Central Kalimantan in 2015, and 75,014 from Kalimantan and Sumatra. See Supplementary Notes 3 and Supplementary Table 1 for full calculations from the broader regions. Further analysis using an alternative methodology which attributes deaths to specific causes by Crippa et al.[28] is also provided in Supplementary Notes 3 and sums of these show generally good agreement with the calculations using the ref. 97 method.

The World Health Organisation estimate that a 24-h exposure to PM$_{2.5}$ concentrations of 75 μg . m$^{-3}$ would lead to around a 5% increase in short term mortality, whilst an annual mean exposure of 35 μg . m$^{-3}$ would lead to a 15% increase in long term mortality[90]. Whilst we do not have reliable in situ measurements for the entire year, the annual mean PM$_{2.5}$ mass concentration for the region can be estimated from the CAMS model outputs as 56 μg . m$^{-3}$. Our in situ PM$_{2.5}$ measurements from the fire season itself,

**Table 3 | Annual mortality estimates for the Palangka Raya area attributable to landscape fire smoke exposure**

| Year | Estimated Annual Deaths in Exposure Cell | Background non-fire PM$_{2.5}$ Concentration ($\mu g \cdot m^{-3}$) | Annual Average PM$_{2.5}$ concentration ($\mu g \cdot m^{-3}$) | Smoke specific PM$_{2.5}$ concentration ($\mu g \cdot m^{-3}$) | Chronically affected attributable mortality |
|---|---|---|---|---|---|
| 2003 | 4106 | 17.9 | 33.84 | 15.94 | 419 |
| 2004 | 4121 | 11.63 | 41.84 | 30.21 | 797 |
| 2005 | 4170 | 10.28 | 30.77 | 20.49 | 547 |
| 2006 | 4078 | 11.6 | 78.21 | 66.61 | 1305 |
| 2007 | 4069 | 9.87 | 15.37 | 5.5 | 143 |
| 2008 | 4049 | 12.54 | 14.21 | 1.67 | 43 |
| 2009 | 4044 | 14.35 | 57.7 | 43.35 | 1122 |
| 2010 | 4019 | 10.01 | 9.59 | −0.42[a] | −11 |
| 2011 | 4063 | 11.54 | 30.48 | 18.94 | 492 |
| 2012 | 4060 | 12.2 | 31.2 | 19 | 494 |
| 2013 | 4112 | 15.85 | 23.65 | 7.8 | 205 |
| 2014 | 4119 | 10.28 | 59.2 | 48.92 | 1290 |
| 2015 | 4190 | 10.83 | 91.87 | 81.04[b] | 1341 |
| 2016 | 4240 | 9.35 | 11.22 | 1.87 | 51 |
| 2017 | 4287 | 9.04 | 11.45 | 2.42 | 66 |
| 2018 | 4379 | 12.14 | 19.99 | 7.85 | 220 |
| 2019 | 4391 | 13.03 | 56.56 | 43.56 | 1223 |

Values are used to calculate the final annual mortality estimate (last column) using the method of ref. 97 and ref. 41. Annual number of deaths in exposure cell are calculated from population estimate for exposure cell, and World Bank death rate (mean 2015–2018) for Indonesia. Modelled airborne PM$_{2.5}$ mass concentrations are from CAMS EAC4 dataset.
[a]Calculation is negative as lack of fire activity means annual average is lower than non-fire season.
[b]This is capped at 50 $\mu g \cdot m^{-3}$ in the mortality calculation[97].

combined with an (unrealistically low) assumption of zero exposure to PM$_{2.5}$ outside of the fire season, provides a mean annual exposure of 22.6 $\mu g \cdot m^{-3}$. It is therefore reasonable to assume that annual exposure of the Palangka Raya population to PM$_{2.5}$ is likely to be above WHO Interim target-1 (35 $\mu g \cdot m^{-3}$) and therefore beyond the scope of most studies that have correlated PM exposure and mortality[91]. This highlights the need for further studies to better understand the impact of higher exposure concentrations and mortality impacts. It is also noted that this lack of understanding may be affecting the estimated health impacts calculated within this study.

In addition to mortality other studies have focussed on assessing hospitalisations, with ref. 101 finding a mean from a number of studies of a 0.25% increase per 1 $\mu g \cdot m^{-3}$ in risk of hospitalisation for respiratory issues from short term (same day) exposure. For Palangka Raya in 2019, this results in a 10.9% increase attributable to fire smoke. An important recognition coming from our PA data is that outdoor air quality is actually far worse overnight than in the day, even though the fires are less active overnight. This means that rather than opening windows at night when it appears that fire activity might have died down somewhat, it might be best for the local populace to keep windows and other openings closed at night but open in the morning to help pollutants diffused into the house overnight dissipate more quickly. Those in higher-quality housing with more sealed windows and better air purification may be better protected from the elevated PM$_{2.5}$ concentrations overnight, and it may conversely make sense for them to keep windows and doors continuously closed. Daily activities (e.g. travel to work or school) may be best moved to later in the morning when concentrations typically reduce. The different methodologies for mortality estimation used in this study provide some variability in specific outputs, however, it is clear that severely degraded AQ from fire smoke, as shown from in-situ and the CAMS model, has a substantial impact on mortality within the region with a wider impact on population health.

## Methods
### Purple Air devices and field deployment
Each of the locations shown in Fig. 1 where the PM measurements were made had a low-cost 'Purple Air' sensor (Model PA-II-SD) installed for the August to October 2019 period. PA sensors were selected for use on the basis of cost

(around $250 at the time of writing) and performance—with them showing amongst the best performance of any low-cost PM$_{2.5}$ sensor during in a series of independent intercomparison studies[55,58,67]. To provide resilience, each Purple Air (PA) device contains two Plantower PMS5003 particulate detectors (termed A and B) that each work on light backscattering principles (Plantower: Beijing, China). Both the A and B detectors output size fractions of PM$_1$, PM$_{2.5}$, PM$_{10}$ along with air temperature and humidity data, and each PA can transmit these data across a Wi-Fi network (Firmware v4.02) whilst also saving it to an internal SD card as backup. Two mass concentrations are actually stored for each PM size class coming from the A and the B sensors, one derived using particulate density and other assumptions appropriate for indoor PM sources, and the other more typical of general outdoor PM sources. We used the latter data, but importantly, we derived a calibration adjustment to account for the lower density of smoke particulates compared to most other outdoor PM (e.g. mineral dust). This re-calibration procedure followed by Delp and Singer.[58] and Mehadi et al.[67]. PA devices are known to slightly overestimate air temperature and relative humidity due to the measurement probes being set quite far inside the instrument case, and so the appropriate adjustment offsets were applied to these data based on Purple Air[102].

We installed ten Purple Air sensors (ID.1 to ID.10) towards the start of the fire season on Kalimantan, during the period 19–31 August 2019. Installation locations spanned the city of Palangka Raya and are shown in Fig. 2, each located 2.5–4 m above ground and as far away from very local sources of particulates as possible (e.g. roads and areas where people congregate to smoke cigarettes). All but one location was at or very close to a Primary Health Care Centre or Midwifery Clinic, giving a spatial sampling pattern broadly reflecting the city's population distribution and enabling the measurement locations to be of maximum relevance to a parallel health impact study then underway[103]. The additional location used for the PA deployment was the Borough Office at Jekan Raya, where a long-term AQMS is also located—operated by the Palangka Raya City Environmental Department. This station comprises a Trusur AQMS that included sensors for particulate matter (PM$_{2.5}$, PM$_{10}$), trace gases (CO, SO$_2$, O$_3$, NO$_2$) as well as meteorology (though not all were working for the full duration of the 2019 fire season, and hence not all are reported here). Each PA logged data almost continuously until their removal on 25 October 2019 after cessation of the fire season.

## Calibration and co-location of Purple Air PM sensors

The two Plantower PMS5003 PM detectors present within each Purple Air sensor are reported by the manufacturer to measure optimally below ~500 µg.m$^{-3}$, with a maximum upper range of ~1000 µg.m$^{-3}$. Based on Wooster et al.[11], such high concentrations were potentially expected during deployment in Palangka Raya during the fire season, so each PA was subject to laboratory testing in a high PM environment to understand its performance under such extreme conditions. We created such high concentrations by burning tropical peat samples in the King's Wildfire Testing Chamber, with further details provided in Supplementary Notes 2. In addition to this laboratory testing, prior to their deployment across Palangka Raya, all PA sensors were subject to colocation testing in situ by placing them in a line 10 cm apart at the Indonesian Bureau of Meteorology, Climatology and Geophysics (BMKG) located at Palangka Raya Airport meteorological station, whose location is also shown in Fig. 2. Colocation tests occurred for 24-h windows between 15th and 18th August 2019, and their purpose was to check for differences in each sensors response to smoke. Simultaneous reference data on absolute airborne PM$_{2.5}$ mass concentrations was also collected during this period to help generate a calibration factor found necessary to adjust the PA output for the inappropriate use of default assumptions when converting light backscattering values into PM mass concentrations (see ref. 11). A gravimetric PM measurement from the tapered element oscillating microbalance (TEOM) normally operating at the BMKG site would be ideal as the reference source, but unfortunately this instrument was inoperable at the time of our co-location tests. Instead, we used a gravimetric method based on pre-conditioned, pre-weighed Teflon coated quartz filters placed in MiniVol filter samplers (Airmetrics MiniVol TAS) that took in air at a rate of 5 litres . min$^{-1}$ for 24 h. MiniVol samplers have routinely provided PM reference data[53,77], with the primary disadvantage compared to the TEOM that only a final 24-h total is provided rather than data on airborne PM concentration variations over the measurement period. For the co-location tests, all Purple Airs were installed in a line 1 m above ground close to the MiniVol inlet, which itself was fitted with a PM$_{2.5}$ inlet to collect data matching the Purple Air PM$_{2.5}$ size fraction. Multiple 24 h mean mass concentration measures were provided by the MiniVols across several days, with each Teflon filter pre-conditioned under controlled conditions of temperature and humidity and weighed pre- and post-exposure using a Mettler Toledo MX5 balance at the National Nuclear Energy Agency of Indonesia[104]. Post-deployment, all PA sensors were again brought to the same co-location site for a period of 9 h on 27th October 2019 to check for any changes in performance related to the extreme PM exposure they received during September 2019 in particular.

Subsequent to the pre-deployment testing, data from each PA installed during the deployment were aggregated using a script written in the R software. This script (i) converted data from UTC to local time, (ii) applied the appropriate PA offset adjustment for temperature and humidity, and (iii) compared the PM values recorded by the Plantower PMS5003 A and B detectors in each unit. If mass concentration differences between the two sensors were <10% for each size fraction, then the mean concentration was calculated for each. Larger differences resulted in the PM concentration measures coming from just the A or B detector, based on which was deemed best performing during the co-location tests. PM data above the 500 µg.m$^{-3}$ were flagged for careful analysis, and readings taken when relative humidity exceeded 75% (16% of measurements) were also flagged and given a higher uncertainty following the work of ref. 64 who demonstrated the effect of such conditions on sensor performance.

## CAMS model analysis

For the comparison to CAMS, the data is compared to both the EAC4 reanalysis dataset[33] as well as the near-real time dataset[40]. These CAMS data come from ingestion into the ECMWF Integrated Forecast System (IFS[69]) of anthropogenic aerosol and pre-cursor emissions from the MACCity and CAMS-GLOB-ANT emissions inventories (covering transportation, energy, industries, ships, residential, solvents and agricultural activities[70,71],

as well as landscape-fire emissions from the Global Fire Assimilation System[34]). It should be noted however, that the model output does not differentiate between fire-derived PM$_{2.5}$ component and that from other sources. Consideration is also given for secondary organic aerosol formation[33]. The current CAMS IFS cycle operates with 137 vertical levels (surface up to 0.01 hPa) at a nominal horizontal resolution of ~40 km. Modelled data outputs are resampled up to 0.125° resolution using the Meteorological Interpolation and Re-gridding scheme[105].

## Attributable mortality estimates

Estimates of excess mortality resulting from chronic exposure to the elevated PM$_{2.5}$ concentrations were calculated using the method of Johnston et al.[97], where excess attributable mortality ($C$) is calculated by:

$$C = M \, x\big([RR_{ci}(PM - CF)] - 1\big) \qquad (1)$$

Where $M$ is number of expected deaths in the exposure cell, $RR_{ci}$ is the relative rate of all-cause mortality per 1 µg . m$^{-3}$ increase in PM$_{2.5}$, $PM$ is the smoke specific annual mean PM$_{2.5}$ concentration (capped at a maximum of 50 µg . m$^{-3}$ according to ref. 97, which is based on exposure limitations of the original study deriving $RR_{ci}$[106]) and $CF$ is the counterfactual PM$_{2.5}$ concentration (which would occur if there were no people lighting fires).

## Data availability

Raw data from the Purple Air sensor network are available from https://doi.org/10.18742/25533376.

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

## Acknowledgements
Fieldwork for this study was supported primarily by UKRI National Capability Official Development Assistance (ST/S003029/1) and by UKRI National Capability funding to the NERC National Centre for Earth Observation (NE/R000115/1 and NE/R016518/1). M.J.G. and M.J.W. are also supported by the Leverhulme Centre for Wildfires, Environment and Society through the Leverhulme Trust (Grant RC-2018-023). V.A. is supported through the Indonesia Endowment Funds for Education (LPDP) and Health Polytechnic of Palangka Raya. P.L. and W.S. are supported by the Ministry of Education (Indonesia) under the World Class University programme managed by Institut Teknologi Bandung (ITB). We thank all staff at Primary Health and Midwifery Centres who enabled the sensor deployment, as well as those who diligently retrieved data from the sensor network. We also thank the anonymous reviewers for their constructive feedback on the manuscript.

## Author contributions
Martin Wooster, Vissia Ardiyani, Mark Grosvenor, Puji Lestari, David Green designed the research. Mark Grosvenor, Vissia Ardiyani, Martin Wooster, Stefan Gillott, Puji Lestari, Wiranda Suri carried out the deployment and data collection, Mark Grosvenor and Vissia Ardiyani did the analysis, Mark Grosvenor, Vissia Ardiyani and Martin Wooster wrote the paper with contributions from all coauthors.

## Competing interests
The authors declare no competing interests.
