## [Transparent Peer Review file · Communications Earth & Environment]

Catastrophic impact of extreme 2019 Indonesian peatland fires on urban air quality and health

Corresponding Author: Dr Mark Grosvenor

Version 0:

Decision Letter:

Dear Dr Grosvenor,

Your manuscript titled "Catastrophic Impact of Extreme Indonesian Peatland Fires on Urban Air Quality and Health Assessed Using a PM_{2.5} Sensor Network and the Copernicus Atmosphere Monitoring System" has now been seen by 3 reviewers, and we include their comments at the end of this message. They find your work of interest, but some important points are raised. We are interested in the possibility of publishing your study in Communications Earth & Environment, but would like to consider your responses to these concerns and assess a revised manuscript before we make a final decision on publication. Please pay particular attention in meeting the following editorial thresholds:

- Explicitly mention the novelty of the research in abstract and discussion
- Clearly discuss the limitations of the results and data analysis

We therefore invite you to revise and resubmit your manuscript, along with a point-by-point response that takes into account the points raised. Please highlight all changes in the manuscript text file.

Please use the following link to submit your revised manuscript, point-by-point response to the referees' comments (which should be in a separate document to any cover letter), a tracked-changes version of the manuscript (as a PDF file) and the completed checklist:

Link Redacted

We hope to receive your revised paper within six weeks; please let us know if you aren't able to submit it within this time so that we can discuss how best to proceed. If we don't hear from you, and the revision process takes significantly longer, we may close your file. In this event, we will still be happy to reconsider your paper at a later date, as long as nothing similar has been accepted for publication at Communications Earth & Environment or published elsewhere in the meantime.

Please do not hesitate to contact us if you have any questions or would like to discuss these revisions further. We look forward to seeing the revised manuscript and thank you for the opportunity to review your work.

Best regards,

Sagar Parajuli, PhD
Editorial Board Member
Communications Earth & Environment

Joe Aslin
Deputy Editor
Communications Earth & Environment

EDITORIAL POLICIES AND FORMATTING

Editorial Policy: [Policy requirements](https://www.nature.com/documents/nr-editorial-policy-checklist.pdf) (Download the link to your computer as a PDF.)

- Behavioural and social science
- Ecological, evolutionary & environmental sciences
- Life sciences

<https://www.nature.com/documents/nr-reporting-summary.zip>

Furthermore, please align your manuscript with our format requirements, which are summarized on the following checklist: [Communications Earth & Environment formatting checklist](https://www.nature.com/documents/commsj-phys-style-formatting-checklist-article.pdf)

and also in our style and formatting guide [Communications Earth & Environment formatting guide](https://www.nature.com/documents/commsj-phys-style-formatting-guide-accept.pdf) .

*** DATA: Communications Earth & Environment endorses the principles of the Enabling FAIR data project (<http://www.copdess.org/enabling-fair-data-project/>). We ask authors to make the data that support their conclusions available in permanent, publically accessible data repositories. (Please contact the editor if you are unable to make your data available).

All Communications Earth & Environment manuscripts must include a section titled "Data Availability" at the end of the Methods section or main text (if no Methods). More information on this policy, is available at <http://www.nature.com/authors/policies/data/data-availability-statements-data-citations.pdf>.

If a community resource is unavailable, data can be submitted to generalist repositories such as [figshare](https://figshare.com/) or [Dryad Digital Repository](http://datadryad.org/). Please provide a unique identifier for the data (for example a DOI or a permanent URL) in the data availability statement, if possible. If the repository does not provide identifiers, we encourage authors to supply the search terms that will return the data. For data that have been obtained from publically available sources, please provide a URL and the specific data product name in the data availability statement. Data with a DOI should be further cited in the methods reference section.

REVIEWER COMMENTS:

Reviewer #1 (Remarks to the Author):

This paper shows measurements of PM2.5 taken in an urban region of Kalimantan during the extreme peatland fires in

2019. Peat fires are known to cause extreme air quality in Indonesia and quantifying this using measurements is important. Low cost air quality monitors were used and have been compared with simulated PM_{2.5} data from the CAMS model, which was then used to estimate the mortality from exposure to fire smoke.

One concern I have with this paper is that the focus and novelty is not made clear. Several previous studies have looked at the health impact from peat fires in Indonesia (some referenced in the paper, some referenced in my comments below). There could be novelty in taking PM_{2.5} measurements and using these to evaluate the model data before using it, or in the work that went into correcting the measurements for the high values seen during the Indonesian fires - these results in particular could be beneficial to future studies, particularly if the low cost sensors haven't been used in this region before.

There is a lot of detail on the calibration of the sensors in the results section. It is important that this calibration was done and the sensors were evaluated, however, the details might be better placed in the supplement to give more emphasis on the air quality results in the main paper. This is only a suggestion, however, and it is up to the authors where they feel it is best placed.

Other specific comments are given below.

Page 2, Line 74: There is a typo: "large-cale"

Page 4, line 146: There is a typo: "his harmful air pollutant"

Page 4, line 162: This phrasing with the sentence before suggests that fire sourced PM_{2.5} has been found more toxic than cigarettes – is this correct?

Page 5, line 185: "really" is not needed with "extreme".

Page 5, line 195: If the exposure is happening on an annual basis how can change in mean annual PM_{2.5} be used to determine a relationship?

Page 6, line 209: The link between the low FRP and the smouldering peat fires could be better explained here.

Figure 3b: From the legend the burned area is red but so is the top end of the FRP scale for the hotspots, so it is unclear what the red is showing.

Page 8, line 282: What are the reasons for data not being captured? Could it be cutting out at high PM values?

Page 8, line 285: I think this sentence needs rewording to make sense: "While the overall temporal pattern that is similar across all sensors, there is a substantial amount of inter-site variability - with the highest concentrations occurring 14 – 16th 286 September when 3-hr averages exceeded 700 µg.m⁻³ 287 at all locations". Possibly "that" shouldn't be in it?

Page 8, line 289: Possible local sinks of PM are mentioned but what about local sources?

Page 9, line 310-311: The phrasing here sounds like the sensors did not properly record the air pollution event, but I think the authors mean that the event occurred before the measurement period?

Page 10, line 314: Can the WHO annual guidelines be given here?

Page 10, line 337: Can the authors comment on what the difference is likely to be between the PM_{2.5} and PM₁₀? If the PM is majority from peat fires, how do PM_{2.5} and PM₁₀ emissions compare?

Figure 7a: Are the number labels for the locations here the ID numbers for the different sensors? So the location is the location of that sensor? I don't think this is really explained.

Page 12, line 366: Stating that urban PM sources are dominant during periods of 'good' air quality could be suggesting that urban PM sources are not a problem for air quality, which is not true.

Page 12, line 370: I think PM_{2.5} concentrations coming from the CAMS model are simulated, not measured?

Page 12, line 373: How large is the city of Palangka Raya? How much of the 168 x 168 km grid (size of 4 84 km grid cells) would be considered city and how much would be outside the city? Could this be affecting the comparison?

Page 12, line 389: With such large differences seen across previous studies, what is the benefit of comparing CAMS data with measurements? Do the other studies give possible reasons for poor comparisons?

Page 12, line 390: This definitely seems like it could be causing the higher model values. I don't know what the location of the city and the fires is to the CAMS grid cells, but is it possible to compare with only the CAMS grid cells which do not contain fire locations?

Page 12, line 398: 'rather well' is a bit vague, is it possible to quantify this?

Page 13, line 407: Again, is it possible to see if the peak occurred in the grid cells on the other side of the city?

Page 13, line 413: 'number' is repeated in the figure caption.

Page 13, line 415: I'm not sure I see the purpose of comparing with the second set of CAMS data?

Page 14, line 437: Can the authors explain here why the PM is capped at 50?

Page 14, line 441: I'm a little confused why a CF of zero is used, as this suggests that there is no PM2.5 from other sources. It is unclear if the CAMS model has been run with all source emissions or only fire emissions and this needs to be described here. If the CAMS model has been run with only fire source emissions then this needs to be mentioned when discussing the comparison between the Purple Air measurements and the CAMS simulations. In Table 3 the non-fire PM2.5 concentrations are given but it is unclear how they are used.

Page 14, table 3: It's unclear what 'Estimated annual deaths in exposure cell' and 'Chronically effected attributable mortality' are. Are either of these the increased mortality caused by fire PM? Or the baseline mortality for the region?

Page 14, line 453: When expanding the study area was PM data from CAMS still used? 'Expanding our methods' could perhaps have a little more description here.

Page 14, line 455: I can't see where the 4910 and 75014 values come from?

It could be worth comparing with other studies which have calculated increased mortality from fire PM2.5.

Edit: I see this has been done in the supplement, but only the Koplitz et al. study has been used. Other studies are:

Crippa P, Castruccio S, Archer-Nicholls S, Lebron G B, Kuwata M, Thota A, Sumin S, Butt E, Wiedinmyer C and Spracklen D V 2016 Population exposure to hazardous air quality due to the 2015 fires in Equatorial Asia Sci. Rep. 6 37074
Kiely, L. et al. Air quality and health impacts of vegetation and peat fires in Equatorial Asia during 2004–2015. Environ. Res. Lett. 15, 1–12 (2020).

Page 14, line 465: What does it mean for the PM concentrations to be outside of the scope? How does this effect the conclusions?

Page 16, line 536: How many readings were taken with humidity greater than 75%? How does a higher uncertainty effect how a reading is used in the overall results of this paper?

Page 16, 538: I think some more details on the CAMS PM2.5 data could be provided in the methods section. Does the model contain emissions from all sources or only fire? Is secondary aerosol formation considered?

Figure S2: There is no description of the purple markers in the figure caption.

Supplement, line 849: The comparison of the PM2.5 concentrations and the active fire data is interesting, but doesn't seem to be referred to from the main paper?

Table S1: Again, I'm not really sure where the non-fire PM2.5 concentrations are coming from. Are these given from the CAMS model?

Reviewer #2 (Remarks to the Author):

This communication piece is very insightful and well written with appropriate attributions and generally robust methodology. It highlights an important topic and interesting findings. I would recommend it for publication pending minor revisions. Please see below for some comments and some suggestions.

--line 62: could be beneficial to insert short mention about other key sources of PM in this region other than peatland fires, typical concentration in non-fire times, etc. while also understanding that PM concentrations are not well understood in this region during that time.

Overall there is opportunity to tie in and highlight related pollution underestimation from the coarse satellite observations over the region, possibly with related AOD ground stations as well (i.e. Palangkarya site).

For the relationship between birth wright and mean annual PM2.5, it is certainly interesting. Authors should also mention about other internal and external factors which can influence the birthweight. I would also wonder how the relationship

would look with a time lagged correlation considering the length of pregnancy which, for many, would extend between multiple years. I would also be interested to see the values other than just the 2015-2019 time period of extreme pollution. This analysis could be more intensive through other approaches, however, the authors convey their main point well.

For land cover map and associated attribution of VIIRS active fire points to land cover, authors should caveat that the data used from the study may have some temporal variation / mismatch and the land classes proportions may be slightly different through time.

Although the study's sensors are located fairly close to the source of fire pollution, there can be other biomass burning sources affecting the measured values. Have the authors checked any wind speed / trajectory data (i.e. MERRA-2), and/or viewed VIIRS active fires-FRP or burned areas to identify potential other sources of pollution from neighboring areas or areas that may be within dispersion range? Results could also be interesting with an air quality transport and dispersion model for comparison.

Even if authors do not do any additional analysis, it would at least be valuable to provide insight on fire counts, FRP, burned area, or GFED emissions across a broader geographic area which would enhance context for the Palangka Raya fire season dynamics.

It would be interesting to see how well GFED or another emissions database does at capturing temporal variation in emissions over peatland fire areas (e.g. another factor in figure 8).

Figure 5 is very informative, however, authors should elaborate more upon the large absolute value differences between AQMS and Purple Air sensors, especially at high AQMS concentration levels. The authors documented an appropriate calibration of the sensors, however, anything that can be done to improve that in the future with calibration or other sensors should be mentioned.

For figure 8 and the associated interpretation in the text: How much of the agreement/disagreement between Purple Air sensors and CAMS is likely attributed to the spatial discrepancies? I would have expected higher concentrations from the in situ sensors than the average of 4 0.75degree grid cells which likely include other areas with lower PM2.5 concentrations. Can the authors elaborate on why they think that is?

The study should consider to explicitly mention shortcomings/uncertainties/limitations of the study.

Reviewer #3 (Remarks to the Author):

Overall comments: I found this paper to be relatively easy to understand, and overall, well-written. The figures greatly added to the paper, and I believe were all relevant. My biggest comment would be to emphasize the importance of both the research and the findings. Make it clear to the reader as to why they should care. Additionally, there was some informality in the language (ex. Using "we" or "us" throughout), though that could be a personal stylistic choice. The results are relevant, and overall, I think it would be a good addition to the current literature.

Introduction:

General: well-written introduction, I feel it sufficiently covers a relevant background to the topic. I think it could use some clarity on the importance of the work, and the specific objectives being addressed in the research. Some repetitiveness.

L40, P1: Run-on sentence, just separate into two – I would do so at the "and by burning" portion of the sentence.

L90, page 3: This sentence is awkwardly phrased.

L90-L109: What you're doing in the paper is very clear, and I think this paragraph is relevant and well-written, but it reads more as a methods section rather than an objectives paragraph. Perhaps emphasizing what the objective/research question of this paper is, followed by a clear sentence on what the "why" is, would provide some clarity to the reader as to the importance of the work, rather than relying on them to deduce it on their own.

Results and Discussion:

General comments: The figures are very helpful, and the results are overall well written. Just some table formatting issues (which may have been a result of the Word document pdf production)

L210, p6 – It's unnecessary to say "fieldwork collected by us", just say "data collected in September 2019..." or something similar.

Table 1 – Edit the table such that the first column is legible (the purple air sensor) formatting is confusing.

Figures 4 & 5 are pretty clear, with no changes to comment on, other than for Figure 5a, the 0-400-800 scale on each ID graph is just a little compressed – maybe reducing the scale font size slightly would make it look less crowded?

Table 2 – Formatting issues with the table column sizes – try to keep the words whole and not separated onto different lines.

** Visit Nature Research's author and referees' website at www.nature.com/authors for information about policies, services and author

benefits**

Communications Earth & Environment is committed to improving transparency in authorship. As part of our efforts in this direction, we are now requesting that all authors identified as 'corresponding author' create and link their Open Researcher and Contributor Identifier (ORCID) with their account on the Manuscript Tracking System prior to acceptance. ORCID helps the scientific community achieve unambiguous attribution of all scholarly contributions. You can create and link your ORCID from the home page of the Manuscript Tracking System by clicking on 'Modify my Springer Nature account' and following the instructions in the link below. Please also inform all co-authors that they can add their ORCIDs to their accounts and that they must do so prior to acceptance.

Version 1:

Decision Letter:

Dear Dr Grosvenor,

Please allow us to apologise for the delay in sending a decision on your manuscript titled "Catastrophic Impact of Extreme Indonesian Peatland Fires on Urban Air Quality and Health Assessed Using a PM_{2.5} Sensor Network and the Copernicus Atmosphere Monitoring System". It has now been seen by our reviewers, whose comments appear below. In light of their advice we are delighted to say that we are happy, in principle, to publish a suitably revised version in Communications Earth & Environment.

We therefore invite you to revise your paper one last time to address the remaining concerns of our reviewers. At the same time we ask that you edit your manuscript to comply with our format requirements and to maximise the accessibility and therefore the impact of your work.

EDITORIAL REQUESTS:

- Please note that we only allow a maximum of 10 display items including figures and tables, in the main text. We request that you move figure 2 to the Supplementary Information file as it is not strictly related to the main result.
- We also suggest that Equation (1) and associated description of excess mortality could be put in the methods section rather in results and discussion.
- Please review our additional specific editorial comments and requests regarding your manuscript in the attached "Editorial Requests Table".

*****Please take care to match our formatting and policy requirements. We will check revised manuscript and return manuscripts that do not comply. Such requests will lead to delays. *****

SUBMISSION INFORMATION:

OPEN ACCESS:

Communications Earth & Environment is a fully open access journal. Articles are made freely accessible on publication. For further information about article processing charges, open access funding, and advice and support from Nature Research, please visit <https://www.nature.com/commsenv/open-access>

At acceptance, you will be provided with instructions for completing the open access licence agreement on behalf of all authors. This grants us the necessary permissions to publish your paper. Additionally, you will be asked to declare that all required third party permissions have been obtained, and to provide billing information in order to pay the article-processing

charge (APC).

Link Redacted

Best regards,

Sagar Parajuli, PhD
Editorial Board Member
Communications Earth & Environment

Joe Aslin
Deputy Editor,
Communications Earth & Environment
<https://www.nature.com/commsenv/>
Twitter: @CommsEarth

REVIEWERS' COMMENTS:

Reviewer #1 (Remarks to the Author):

I am happy with the majority of the responses to my comments. In particular, I think the edits made to the abstract and discussion mean the importance of the paper is now much clearer.

There are two of my original comments which I do not feel have been fully addressed, explained below.

Line 492: "we estimate excess deaths from landscape fire PM2.5 exposure to be 3276 and 51,377 respectively, which compares to 4910 attributable deaths in Central Kalimantan in 2015, and 75,014 from Kalimantan and Sumatra"
The estimates from this paper are being compared with 4910 and 75014 attributable deaths, but I cannot see where these values have come from. I might be misunderstanding what 'attributable deaths' refers to, and how this differs from 'deaths from landscape fire PM2.5 exposure'.

Line 580: Thank you for informing me that 16% of the readings were flagged for having humidity greater than 75%, and therefore a larger uncertainty. I think it would be good to include this in the text as a reader may want to know how much of your dataset this is reflecting.

Reviewer #2 (Remarks to the Author):

My comments have been addressed; the manuscript is suitable to publish.

COMMSENV-24-0828 Author Responses to Reviewers (responses given in *red italics*)

Reviewer #1 (Remarks to the Author):

This paper shows measurements of PM2.5 taken in an urban region of Kalimantan during the extreme peatland fires in 2019. Peat fires are known to cause extreme air quality in Indonesia and quantifying this using measurements is important. Low cost air quality monitors were used and have been compared with simulated PM2.5 data from the CAMS model, which was then used to estimate the mortality from exposure to fire smoke.

One concern I have with this paper is that the focus and novelty is not made clear. Several previous studies have looked at the health impact from peat fires in Indonesia (some referenced in the paper, some referenced in my comments below). There could be novelty in taking PM2.5 measurements and using these to evaluate the model data before using it, or in the work that went into correcting the measurements for the high values seen during the Indonesian fires - these results in particular could be beneficial to future studies, particularly if the low cost sensors haven't been used in this region before.

Thank you for your detailed comments on our manuscript. We have rephrased part of the abstract to help make the novelty clear (whilst simultaneously avoiding any of the language noted in the journal style guidelines). We believe this study does provide new insight into how a low-cost sensor network can be used in environments such as this, where air quality may be beyond what instruments such as these are typically designed or expected to measure in urban areas. We have also added additional statements in the discussion section to highlight the novelty of the research. (Lines 84-5, 103-5, 256-257, 278-284, 297-298, 376-379, 518-521)

There is a lot of detail on the calibration of the sensors in the results section. It is important that this calibration was done and the sensors were evaluated, however, the details might be better placed in the supplement to give more emphasis on the air quality results in the main paper. This is only a suggestion, however, and it is up to the authors where they feel it is best placed.

Thank you for this suggestion. We have moved some details from the main text to the methods section, however, we feel that many of the points in relation to calibration within the main text are key points which reflect the reliability of our sensor dataset.

Other specific comments are given below.

Page 2, Line 74: There is a typo: "large-cale"

This has been corrected

Page 4, line 146: There is a typo: "his harmful air pollutant"

This has been corrected

Page 4, line 162: This phrasing with the sentence before suggests that fire sourced PM2.5 has been found more toxic than cigarettes – is this correct?

This statement has been clarified with the sentence rephrased to state ‘non-fire’ sources to help differentiate between carbonaceous-derived PM and other PM sources.

Page 5, line 185: “really” is not needed with “extreme”.

This has been corrected.

Page 5, line 195: If the exposure is happening on an annual basis how can change in mean annual PM2.5 be used to determine a relationship?

This statement has been rephrased and a reference to Figure 1 added which highlights that 70% of years between 2003-2019 had a significant fire season.

Page 6, line 209: The link between the low FRP and the smouldering peat fires could be better explained here.

An additional sentence has been added here to help explain why these fires show lower FRP.

Figure 3b: From the legend the burned area is red but so is the top end of the FRP scale for the hotspots, so it is unclear what the red is showing.

Thank you for highlighting this – we have edited the burned area shading to differentiate this

Page 8, line 282: What are the reasons for data not being captured? Could it be cutting out at high PM values?

If the PM sensors are fully saturated or issues with the PM sensor components there would still be some data recorded on the SD Card (eg temperature & humidity). The gaps in data during deployment were due to power being lost to the instrument temporarily. A sentence clarifying this has been added to the paragraph.

Page 8, line 285: I think this sentence needs rewording to make sense: “While the overall temporal pattern that is similar across all sensors, there is a substantial amount of inter-site variability - with the highest concentrations occurring 14 – 16th 286 September when 3-hr averages exceeded 700 $\mu\text{g.m}^{-3}$ 287 at all locations”. Possibly “that” shouldn’t be in it?

This has been corrected.

Page 8, line 289: Possible local sinks of PM are mentioned but what about local sources?

This is a good point that should have been included. The sentence has been updated to mention this.

Page 9, line 310-311: The phrasing here sounds like the sensors did not properly record the air pollution event, but I think the authors mean that the event occurred before the measurement period?

This sentence has been rephrased to make this point clear that it was because the network was not installed, rather than sensor malfunction.

Page 10, line 314: Can the WHO annual guidelines be given here?

This value has now been added.

Page 10, line 337: Can the authors comment on what the difference is likely to be between the PM_{2.5} and PM₁₀? If the PM is majority from peat fires, how do PM_{2.5} and PM₁₀ emissions compare?

Our data shows that around 90% of PM₁₀ concentrations are formed of PM_{2.5} sized particles, although this calculation is based on raw, uncalibrated mass concentrations from the sensors as we have not calculated a PM₁₀ correction factor for peat-fire sources. A sentence has been added to include this information.

Figure 7a: Are the number labels for the locations here the ID numbers for the different sensors? So the location is the location of that sensor? I don't think this is really explained.

Whilst the sensor IDs and location numbers are theoretically separate identifiers, in the case of the deployment the numbers always matched as swapping of sensors was not required. The axis label in Figure 7a has been updated to make this clearer.

Page 12, line 366: Stating that urban PM sources are dominant during periods of 'good' air quality could be suggesting that urban PM sources are not a problem for air quality, which is not true.

This is a good point. The reference to 'good' air quality has now been removed to avoid misinterpretation.

Page 12, line 370: I think PM_{2.5} concentrations coming from the CAMS model are simulated, not measured?

This is correct. The sentence has been rephrased to make this distinction clear.

Page 12, line 373: How large is the city of Palangka Raya? How much of the 168 x 168 km grid (size of 4 84 km grid cells) would be considered city and how much would be outside the city? Could this be effecting the comparison?

The administrative boundary of Palangka Raya covers 9.4% of the area of the CAMS analysis area, however, it is noted that only a proportion of the administrative area is built upon. It is quite right that this discrepancy in spatial area is affecting the comparison between larger scale EO data and localised sensor data. We have added in explanation to the discussion section to address this point more clearly whilst still noting that understanding whether larger scale EO data is valuable for understanding more localised impacts upon a population.

Page 12, line 389: With such large differences seen across previous studies, what is the benefit of comparing CAMS data with measurements? Do the other studies give possible reasons for poor comparisons?

The CAMS model uses a wide range of assumptions about fire emissions and land cover, and therefore its performance is expected to vary in different environments. Knowledge of how well the model is performing in different environments is important to enable targeted improvements to the

model in other areas, and also to provide information on how reliable the model data may be in a particular region.

Page 12, line 390: This definitely seems like it could be causing the higher model values. I don't know what the location of the city and the fires is to the CAMS grid cells, but is it possible to compare with only the CAMS grid cells which do not contain fire locations?

As outlined in the previous comment, it is likely that a substantial amount of $PM_{2.5}$ from the CAMS model is coming from fires within the broader grid cells used for analysis and hence a discrepancy with the in-situ measurements. Whilst it would theoretically be good to compare to grid cells that do not contain fires, this is problematic within this region as fires are widespread in this area of Central Kalimantan and therefore there are not CAMS grid cells available that contain similar landscape and population, but no fires where this could be compared better.

Page 12, line 398: 'rather well' is a bit vague, is it possible to quantify this?

The r^2 agreement for this sub-period has now been calculated and added to the end of the sentence.

Page 13, line 407: Again, is it possible to see if the peak occurred in the grid cells on the other side of the city?

As described in the text, this area of fire activity was within the CAMS grid cells used for analysis of exposure and therefore the hotspot of fire activity will be accounted for in $PM_{2.5}$ estimations for this grid cell. Broader interrogation across the wider region of source attribution of CAMS $PM_{2.5}$ would be interesting, however, we have not expanded our analysis within this paper to keep the focus on assessing the impacts of AQ to a city population. We have, however, included this sentence about hotspot activity not being measured by our sensor network to help acknowledge the range of factors influencing agreement that result from spatial scales of the two datasets.

Page 13, line 413: 'number' is repeated in the figure caption.

This has been corrected.

Page 13, line 415: I'm not sure I see the purpose of comparing with the second set of CAMS data?

Both the near real-time (NRT) and reanalysis (EAC4) datasets are mentioned as there are some differences in how the CAMS model works with these. The lower agreement between the in-situ measurements and the NRT data means that near-real-time use of CAMS data is shown to be slightly less reliable than the reanalysis data. The reanalysis data has the advantage of including data from additional sources which should improve the reliability of the output, however, this data is only useful for retrospective analysis. A clarifying statement has been added to the main text to justify the comparison to the NRT dataset.

Page 14, line 437: Can the authors explain here why the PM is capped at 50?

The $50 \mu\text{g.m}^{-3}$ cap is an assumption by Johnston et al (2012) that the exposure-mortality relationship used is valid up to $50 \mu\text{g.m}^{-3}$. The original study this relationship is based on is from Pope et al (1995) which shows data up to $35 \mu\text{g.m}^{-3}$ for fine particulate matter in deriving their relationship. A statement clarifying this has been added to the paper including reference to the original study.

Pope CA III, Thun MJ, Namboodiri MM, Dockery OW, Evans JS, Speizer FE, Heath Jr CW. Particulate air pollution as a predictor of mortality in a prospective study of U.S. adults. Am J Respir Crit Care Med 1995; 151:669-74

Page 14, line 441: I'm a little confused why a CF of zero is used, as this suggests that there is no PM_{2.5} from other sources. It is unclear if the CAMS model has been run with all source emissions or only fire emissions and this needs to be described here. If the CAMS model has been run with only fire source emissions then this needs to be mentioned when discussing the comparison between the Purple Air measurements and the CAMS simulations.

In Table 3 the non-fire PM_{2.5} concentrations are given but it is unclear how they are used.

The exposure calculation is only to quantify impacts from fire-smoke derived PM_{2.5}, hence the deduction of the non-fire season background signal (as given in Table 3) to account for PM_{2.5} from other non-fire sources which are present during the fire season. A CF of 0 is used as outside of the fire-season in this region there would not naturally be fires within the region. The standard CAMS model outputs used in this study do not differentiate between fire-derived and non-fire derived PM. Background non-fire PM_{2.5} concentrations are used in the equation to remove the impact of exposure throughout the year to other sources of PM_{2.5}. An additional explanation has been added to the paper to better explain this.

Page 14, table 3: It's unclear what 'Estimated annual deaths in exposure cell' and 'Chronically effected attributable mortality' are. Are either of these the increased mortality caused by fire PM? Or the baseline mortality for the region?

The estimated annual deaths in exposure cell is based on the sum of the gridded population within the cell (data based on the UN methodology) multiplied by the World Bank based death rate for Indonesia. This provides an estimate of the total number of all-cause mortality within the exposure cell in each year (to account for any changes in population and changes in death rate). The chronically affected attributable mortality is the number of deaths within that same population that could be attributable to the smoke-derived PM_{2.5}. Additional explanation has been added to the table caption to help clarify the columns.

Page 14, line 453: When expanding the study area was PM data from CAMS still used? 'Expanding our methods' could perhaps have a little more description here.

A reference to this explaining that the expansion still using the same data sources as inputs has been added.

Page 14, line 455: I can't see where the 4910 and 75014 values come from?

It could be worth comparing with other studies which have calculated increased mortality from fire PM_{2.5}.

Edit: I see this has been done in the supplement, but only the Koplitz et al. study has been used.

Other studies are:

Crippa P, Castruccio S, Archer-Nicholls S, Lebron G B, Kuwata M, Thota A, Sumin S, Butt E, Wiedinmyer C and Spracklen D V 2016 Population exposure to hazardous air quality due to the 2015 fires in Equatorial Asia Sci. Rep. 6 37074

Kiely, L. et al. Air quality and health impacts of vegetation and peat fires in Equatorial Asia during 2004–2015. Environ. Res. Lett. 15, 1–12 (2020).

Thank you for this suggestion, we have now added additional analysis to the Supplementary Information section to include mortality estimates for 2015 and 2019 based on methods of Crippa et al (2016) as this enables some insight into the specific diseases which are affected by $PM_{2.5}$. As our methods only consider the CAMS output rather than running a specific model with fire inputs included or removed, the annual mean $PM_{2.5}$ concentrations used for non-fire $PM_{2.5}$ are based on the background concentrations used previously in our analysis. Using the Crippa et al Attributable Fraction equation (Crippa et al, 2016 eqn 6), we apply this to the population areas previously used in our analysis (Indonesia as a whole, Sumatra, Kalimantan, Central Kalimantan, and finally the exposure cell over Palangka Raya). As with the Crippa et al (2016) analysis the Relative Risk rates are taken from Apte et al (2015), whilst disease specific mortality rates are taken from the Global Burden of Disease Mortality estimates for 2019 for the regions considered in this study. The calculations show broad agreement between the sum of all specific diseases using the Crippa et al (2016) methodology, and those calculated for all-cause mortality using the Johnston et al (2012) method. To aid comparisons between the two methods an additional column has been added to Table S1 repeating the Palangka Raya data which is previously given in Table 3 meaning it aligns with columns in Table S2.

Page 14, line 465: What does it mean for the PM concentrations to be outside of the scope? How does this effect the conclusions?

The $PM_{2.5}$ concentrations that have mostly been used in health impacts studies have been lower than what is experienced within this area. There is a need to better understand how these higher concentrations are affecting people's health as relationships between exposure concentrations and health impacts may not be linear. Future research could be targeted to help address this by focusing on extreme levels of exposure over many years. An additional sentence has been added to highlight this conclusion.

Page 16, line 536: How many readings were taken with humidity greater than 75%? How does a higher uncertainty effect how a reading is used in the overall results of this paper?

16% of readings had a relative humidity of > 75%. There was no clear relationship between high relative humidity and high PM concentrations (e.g. these were not only related to high or low PM concentrations). The measurements with high relative humidity were, however flagged during analysis although highlighting these points on data plots where many values were averaged (eg 3 hour mean, or multi-sensor means) meant that it was not possible to clearly show these without making the plots overly complex to interpret.

Page 16, 538: I think some more details on the CAMS $PM_{2.5}$ data could be provided in the methods section. Does the model contain emissions from all sources or only fire? Is secondary aerosol formation considered?

The CAMS model output for $PM_{2.5}$ is a single value which does not discriminate between sources. Secondary organic aerosol formation is included as a factor in the CAMS model. We have moved some of the description of CAMS given in the 'Airborne Particulate Matter – Measurement, Modelling and Health Impacts' section to the Methods section and included these additional details. The full description of the CAMS model is given by Innes et al (2019) and we refer the reader to this for detailed insight into the model.

Figure S2: There is no description of the purple markers in the figure caption.

This has been now been added to the figure caption.

Supplement, line 849: The comparison of the PM_{2.5} concentrations and the active fire data is interesting, but doesn't seem to be referred to from the main paper?

This has now been referred to more explicitly within the discussion of FRP in the Land Cover and Fire Activity section.

Table S1: Again, I'm not really sure where the non-fire PM_{2.5} concentrations are coming from. Are these given from the CAMS model?

As with the calculations for the Palangka Raya exposure cell within the main text, the PM_{2.5} data used in these calculations is from the standard CAMS outputs which are total PM_{2.5} from all sources. The background concentrations are calculated to remove the non-fire derived PM_{2.5} component of this. An additional explanation is given to highlight the methodology for Table S1 is the same as that used in Table 3 within the main paper.

Reviewer #2 (Remarks to the Author):

This communication piece is very insightful and well written with appropriate attributions and generally robust methodology. It highlights an important topic and interesting findings. I would recommend it for publication pending minor revisions. Please see below for some comments and some suggestions.

Thank you for your constructive comments below.

--line 62: could be beneficial to insert short mention about other key sources of PM in this region other than peatland fires, typical concentration in non-fire times, etc. while also understanding that PM concentrations are not well understood in this region during that time.

The fire season mean PM_{2.5} concentration is 97 $\mu\text{g}\cdot\text{m}^{-3}$ (2003-2019, August-October) whilst the non-fire season background mean is 14 $\mu\text{g}\cdot\text{m}^{-3}$ (all other months). We have added this information to this paragraph together with a note about the likely source of non-fire PM_{2.5}.

Overall there is opportunity to tie in and highlight related pollution underestimation from the coarse satellite observations over the region, possibly with related AOD ground stations as well (i.e. Palangkarya site).

Our findings demonstrate that coarse satellite datasets (such as CAMS) can in fact overestimate PM_{2.5} exposure on a population who are located within the analysis cell where fire sources are also present. This highlights the need to consider the location of the population in question along with local fire sources in landscapes such as this where there is not a single fire source/region affecting a population. Further analysis of the relationship with other related measures such as the various AOD data sources would be interesting, but have not been included here as our focus was on how well the CAMS model represents exposure to the population.

For the relationship between birth weight and mean annual PM_{2.5}, it is certainly interesting. Authors should also mention about other internal and external factors which can influence the birthweight. I

would also wonder how the relationship would look with a time lagged correlation considering the length of pregnancy which, for many, would extend between multiple years. I would also be interested to see the values other than just the 2015-2019 time period of extreme pollution. This analysis could be more intensive through other approaches, however, the authors convey their main point well.

We agree these are interesting points that require further in-depth investigation, however, this is beyond the scope of what is possible within this paper. The more precise relationship with individual exposure periods linked to individual birthweights alongside additional epidemiological datasets will be published in a future paper as far more detail is required in order to fully explain these datasets. We include the data in Figure 2 to demonstrate that even at a basic level there appears to be a relationship between PM_{2.5} and birthweight in this area. We investigated a wider range of years for birthweight data, but concluded that the hospital datasets from years prior to 2015 were not recorded systematically enough to be used as a reliable data source enabling inter-year comparisons. There may however, be some more indicative conclusions that could come from this wider data source in future research.

For land cover map and associated attribution of VIIRS active fire points to land cover, authors should caveat that the data used from the study may have some temporal variation / mismatch and the land classes proportions may be slightly different through time.

This is a good point – we have added in a sentence to caveat the explanation of fire locations.

Although the study's sensors are located fairly close to the source of fire pollution, there can be other biomass burning sources affecting the measured values. Have the authors checked any wind speed / trajectory data (i.e. MERRA-2), and/or viewed VIIRS active fires-FRP or burned areas to identify potential other sources of pollution from neighboring areas or areas that may be within dispersion range? Results could also be interesting with an air quality transport and dispersion model for comparison.

Whilst many of the hotspots are located very close to the city, there are many additional hotspots in the wider region. The area covered by the CAMS 4-cell area contains 42% of the total number of hotspots within the wider Central Kalimantan region and 37% of the burned area whilst the analysis area is . This means that there may be sources beyond the CAMS 4-cell area that are impacting AQ in Palangka Raya. This detail has been added to Supplementary Information. Full analysis of wind data to help attribute source area would be interesting, however, we determined that this would benefit from a finer resolution PM dataset than CAMS provides and therefore would be beyond the scope of the aims of this paper.

Even if authors do not do any additional analysis, it would at least be valuable to provide insight on fire counts, FRP, burned area, or GFED emissions across a broader geographic area which would enhance context for the Palangka Raya fire season dynamics.

We have included additional statistics to highlight the active fire data for the wider Central Kalimantan region to the discussion in Supplementary Information.

It would be interesting to see how well GFED or another emissions database does at capturing temporal variation in emissions over peatland fire areas (e.g. another factor in figure 8).

In general GFED follows a broadly similar temporal pattern to the VIIRS data, however, it is notable

that during the early phase of burning, GFED does not show the same degree of temporal variability as VIIRS. We have included the GFED data to Figure S4b as this helps to show the agreement between hotspots, modelled emissions, and modelled PM_{2.5}. Sentences have also been added to the Supplementary Information text discussing the active fire data.

Figure 5 is very informative, however, authors should elaborate more upon the large absolute value differences between AQMS and Purple Air sensors, especially at high AQMS concentration levels. The authors documented an appropriate calibration of the sensors, however, anything that can be done to improve that in the future with calibration or other sensors should be mentioned.

We understand the AQMS was calibrated shortly after our measurement period, and as mentioned in the manuscript was not working during the start of our measurement period therefore it is likely the discrepancy between the Purple Airs and the AQMS is at least in part due to calibration requirements of the AQMS. In any future similar work, a more extended period of co-located measurements between small sensors and fixed stations could aid greatly in characterising PM measurements in a better way, but for this paper it is difficult to say how typical the AQMS data is for that particular instrument in that location. We have added a sentence to the text to note that the AQMS data is indicative and may not be fully representative of that instrument in other years.

For figure 8 and the associated interpretation in the text: How much of the agreement/disagreement between Purple Air sensors and CAMS is likely attributed to the spatial discrepancies? I would have expected higher concentrations from the in situ sensors than the average of 4 0.75 degree grid cells which likely include other areas with lower PM_{2.5} concentrations. Can the authors elaborate on why they think that is?

As with similar comments to Reviewer 1, it is very likely that the discrepancy of spatial scales between the CAMS datasets and sensor datasets are resulting in key differences. As our manuscript notes however, the agreement is generally improved when concentrations are lower. CAMS estimates may be increased compared to what is experienced within the city area as a result of source areas being within a grid cell where PM emissions can be an order of magnitude higher than in 'ambient' air. So whilst the larger area of the CAMS analysis is likely to include areas with lower concentrations, the very high concentrations from fire sources are likely to be increasing the overall estimate.

The study should consider to explicitly mention shortcomings/uncertainties/limitations of the study.

We have added in statements to identify limitations of this study in particular lines 401-404, 461-563, and 505-507, in addition to new statements clarifying other specific points raised by reviewers.

Reviewer #3 (Remarks to the Author):

Overall comments: I found this paper to be relatively easy to understand, and overall, well-written. The figures greatly added to the paper, and I believe were all relevant. My biggest comment would be to emphasize the importance of both the research and the findings. Make it clear to the reader as to why they should care. Additionally, there was some informality in the language (ex. Using "we" or "us" throughout), though that could be a personal stylistic choice. The results are relevant, and

overall, I think it would be a good addition to the current literature.

Thank you for your useful comments and suggestions. As stated in comments to Reviewer 1, we have added in several statements throughout the discussion which emphasise the importance of the research and its findings. In particular line numbers (revised manuscript): 84-5, 103-5, 256-257, 278-284, 297-298, 376-379, 518-521

Introduction:

General: well-written introduction, I feel it sufficiently covers a relevant background to the topic. I think it could use some clarity on the importance of the work, and the specific objectives being addressed in the research. Some repetitiveness.

Following this comment and those from other reviewers various statements throughout the manuscript have been edited or added to help emphasise the importance of the work and make the objectives clear.

L40, P1: Run-on sentence, just separate into two – I would do so at the “and by burning” portion of the sentence.

We agree with this suggestion and have made this edit.

L90, page 3: This sentence is awkwardly phrased.

This sentence has been rephrased to make it clearer.

L90-L109: What you’re doing in the paper is very clear, and I think this paragraph is relevant and well-written, but it reads more as a methods section rather than an objectives paragraph. Perhaps emphasizing what the objective/research question of this paper is, followed by a clear sentence on what the “why” is, would provide some clarity to the reader as to the importance of the work, rather than relying on them to deduce it on their own.

Thank you for this suggestion. We have added in a couple of statements which should make the reasoning for the objectives clearer.

Results and Discussion:

General comments: The figures are very helpful, and the results are overall well written. Just some table formatting issues (which may have been a result of the Word document ⇄ pdf production)

L210, p6 – It’s unnecessary to say “fieldwork collected by us”, just say “data collected in September 2019...” or something similar.

This has been edited.

Table 1 – Edit the table such that the first column is legible (the purple air sensor) formatting is confusing.

This appears to have been caused during the system upload conversion to .pdf format – the original word document does not have this formatting issue.

Figures 4 & 5 are pretty clear, with no changes to comment on, other than for Figure 5a, the 0-400-800 scale on each ID graph is just a little compressed – maybe reducing the scale font size slightly would make it look less crowded?

The font has now been edited to improve spacing.

Table 2 – Formatting issues with the table column sizes – try to keep the words whole and not separated onto different lines.

Again, this appears to be a result of the online system upload as the original table does not have this issue.

COMMSENV-24-0828 Author Final Responses to Reviewers (responses given in *red italics*)

Reviewer #1 (Remarks to the Author):

I am happy with the majority of the responses to my comments. In particular, I think the edits made to the abstract and discussion mean the importance of the paper is now much clearer.

Thank you for your suggestions – these have certainly improved the overall manuscript.

There are two of my original comments which I do not feel have been fully addressed, explained below.

Line 492: "we estimate excess deaths from landscape fire PM2.5 exposure to be 3276 and 51,377 respectively, which compares to 4910 attributable deaths in Central Kalimantan in 2015, and 75,014 from Kalimantan and Sumatra"

The estimates from this paper are being compared with 4910 and 75014 attributable deaths, but I cannot see where these values have come from. I might be misunderstanding what 'attributable deaths' refers to, and how this differs from 'deaths from landscape fire PM2.5 exposure'.

We realise the confusion here likely stems from the phrasing of the sentence – we have clarified in the final manuscript that the first two values (3276 and 51377) are for the 2019 estimates, whilst the latter values are using the same calculation based on 2015 data.

Line 580: Thank you for informing me that 16% of the readings were flagged for having humidity greater than 75%, and therefore a larger uncertainty. I think it would be good to include this in the text as a reader may want to know how much of your dataset this is reflecting.

We have now added this statistic into the methods text.

Reviewer #2 (Remarks to the Author):

My comments have been addressed; the manuscript is suitable to publish.

Thank you for your previous suggestions. We are pleased you now feel this is suitable for publication.

Reviewer #3 (Remarks to the Author):

N/A